# AN INSTANCE-LEVEL FRAMEWORK FOR MULTI-TASKING GRAPH SELF-SUPERVISED LEARNING

## ABSTRACT

With hundreds of graph self-supervised pretext tasks proposed over the past few years, the research community has greatly developed, and the key is no longer to design more powerful but complex pretext tasks, but to make more effective use of those already on hand. There have been some pioneering works, such as AutoSSL (Jin et al., 2021) and ParetoGNN (Ju et al., 2022), proposed to balance multiple pretext tasks by global loss weighting in the pre-training phase. Despite their great successes, several tricky challenges remain: *(i)* they ignore instance-level requirements, i.e., different instances (nodes) may require localized combinations of tasks; *(ii)* poor scalability to emerging tasks, i.e., all task losses need to be re-weighted along with the newly added task and re-pretrained; *(iii)* no theoretical guarantee of benefiting from more tasks, i.e., more tasks do not necessarily lead to better performance. To address the above issues, we propose in this paper a novel multi-teacher knowledge distillation framework for instance-level Multi-tasking Graph Self-Supervised Learning (`MGSSL`), which trains multiple teachers with different pretext tasks, then integrates the knowledge of different teachers *for each instance separately* by two parameterized knowledge integration schemes (`MGSSL-TS` and `MGSSL-LF`), and finally distills it into the student model. Such a framework shifts the trade-off among multiple pretext tasks from loss weighting in the pre-training phase to knowledge integration in the fine-tuning phase, making it compatible with an arbitrary number of pretext tasks without the need to re-pretrain the entire model. Furthermore, we theoretically justify that `MGSSL` has the potential to benefit from a wider range of teachers (tasks). Extensive experiments have shown that by combining a few simple but classical pretext tasks, the resulting performance is comparable to the state-of-the-art competitors.

## 1 INTRODUCTION

Deep learning on graphs (Wu et al., 2020) has recently achieved remarkable success on a variety of tasks, while such success relies heavily on the massive and carefully labeled data. However, precise annotations are usually very expensive and time-consuming. Recent advances in graph *Self-supervised Learning* (SSL) (Wu et al., 2021; Xie et al., 2021; Liu et al., 2021) have provided novel insights into reducing the dependency on annotated labels and enable the training on massive unlabeled data. The primary goal of graph SSL is to provide self-supervision for learning transferable knowledge from abundant unlabeled data, through well-designed pretext tasks (in the form of loss functions). There have been hundreds of pretext tasks proposed in the past few years (Sun et al., 2019; Hu et al., 2019; Xia et al., 2022; 2021; Zhu et al., 2020a; You et al., 2020a; Zhang et al., 2020), and different pretext tasks extract different levels of graph knowledge based on different inductive biases. For example, `PAIRDIS` (Jin et al., 2020) captures the inter-node long-range dependencies by predicting the shortest path lengths between nodes, while `PAR` (You et al., 2020b) extracts topological information by predicting the graph partitions of nodes. With so many ready-to-use pretext tasks already on hand, as opposed to designing more complex pretext tasks, a more promising problem here is *how to leverage multiple existing pretext tasks more effectively*.

There have been some previous works, such as AutoSSL (Jin et al., 2021) and ParetoGNN (Ju et al., 2022), that propose to adaptively weight the losses of different pretext tasks in the pre-training phase with the optimization objective of graph homophily or Pareto optimality. Despite the great progress, there are still several tricky challenges. Firstly, they both ignore instance-level requirements, i.e.,

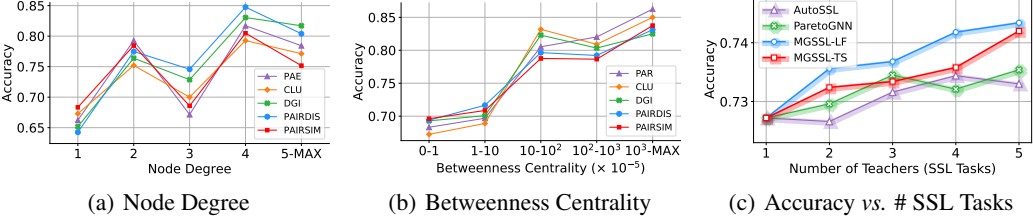

(a) Node Degree      (b) Betweenness Centrality      (c) Accuracy *vs.* # SSL Tasks

Figure 1: *(a)(b)* Classification accuracy of nodes with different node degrees and betweenness centrality across five pretext tasks on the Citeseer dataset. *(c)* Classification accuracy of AutoSSL, ParetoGNN, and MGSSL with respect to the number of SSL tasks on the Citeseer dataset.

different instances (nodes) may require localized and customized combinations of pretext tasks. To illustrate this, we report the classification accuracy of nodes with different node degrees across five pretext tasks (`PAR`, `CLU` (You et al., 2020b), `DGI` (Velickovic et al., 2019), `PAIRDIS` and `PAIRSIM` (Jin et al., 2020)) in Fig. 1(a), from which we observe that different nodes may require localized pretext tasks; for example, high-degree nodes benefit more from `DGI` and `PAIRDIS`, while low-degree nodes prefer `CLU` and `PAIRSIM`. Another example with Betweenness Centrality as a metric in Fig. 1(b) shows the same phenomenon, which calls for an instance-level framework for multi-tasking graph self-supervise. Secondly, balancing multiple tasks by loss weighting during the pre-training phase makes it hard to scale the pre-trained model to emerging tasks. To incorporate new tasks, it requires to re-weight the losses of new tasks and existing tasks to re-pretrain the model. Finally, we present the performance of AutoSSL, ParetoGNN, and MGSSL as the number of SSL tasks increases in Fig. 1(c), which shows that only MGSSL can consistently benefit from more tasks.

**Present Work.** To address the above issues, this paper proposes a novel multi-teacher knowledge distillation framework for instance-level Multi-tasking Graph SSL (`MGSSL`), which trains multiple teachers with different pretext tasks and then integrates the knowledge of different teachers for each instance separately by two parameterized knowledge integration schemes (`MGSSL-TS` and `MGSSL-LF`). This framework shifts the trade-off among multiple pretext tasks from loss weighting in the pre-training phase to knowledge integration in the fine-tuning phase. As a result, when a new task is encountered, we no longer need to re-weight all task losses for pre-training, but simply train a model with only the new task and use it as an additional teacher for knowledge integration, and finally distill the integrated knowledge into the student model. Furthermore, we provide a provable theoretical guideline for how to integrate the knowledge of different teachers, i.e., the integrated teacher probability should be close to the true class-Bayesian probability. More importantly, we prove theoretically that the optimal integrated teacher probability can monotonically approach the Bayesian class-probability as the number of teachers (SSL tasks) increases, which demonstrates that `MGSSL` has the theoretical potential to benefit from a wider range of teachers (SSL tasks). Extensive experiments on eight graph datasets have shown that by combining a few simple but classical pretext tasks, the resulting performance of `MGSSL` is comparable to that of state-of-the-art competitors.

## 2 Preliminaries

**Notations.** Let $\mathcal{G} = (\mathcal{V}, \mathcal{E}, \mathbf{X})$ denote an attributed graph, where $\mathcal{V}$ is the set of $|\mathcal{V}| = N$ nodes with features $\mathbf{X} = [\mathbf{x}_1, \mathbf{x}_2, \cdots, \mathbf{x}_N] \in \mathbb{R}^{N \times d}$ and $\mathcal{E} \subseteq \mathcal{V} \times \mathcal{V}$ is the set of $|\mathcal{E}|$ edges between nodes. Following the common semi-supervised node classification setting, only a subset of node $\mathcal{V}_L = \{v_1, v_2, \cdots, v_L\}$ with corresponding labels $\mathcal{Y}_L = \{y_1, y_2, \cdots, y_L\}$ are known, and we denote the labeled set as $\mathcal{D}_L = (\mathcal{V}_L, \mathcal{Y}_L)$ and unlabeled set as $\mathcal{D}_U = (\mathcal{V}_U, \mathcal{Y}_U)$, where $\mathcal{V}_U = \mathcal{V} \backslash \mathcal{V}_L$. The task of node classification aims to learn a GNN encoder $f_\theta(\cdot)$ and a linear prediction head $g_\omega(\cdot)$ with the task loss $\mathcal{L}_{\text{task}}(\theta, \omega)$ on labeled data $\mathcal{D}_L$, so that they can be used to infer the labels $\mathcal{Y}_U$.

**Problem Statement.** Given a GNN encoder $f_\theta(\cdot)$, a prediction head $g_\omega(\cdot)$, and $K$ losses of self-supervised tasks $\{\mathcal{L}_{\text{ssl}}^{(1)}(\theta, \eta_1), \mathcal{L}_{\text{ssl}}^{(2)}(\theta, \eta_2), \cdots, \mathcal{L}_{\text{ssl}}^{(K)}(\theta, \eta_K)\}$ with prediction heads $\{g_{\eta_k}(\cdot)\}_{k=1}^K$, two common strategies for combining self-supervised task losses $\{\mathcal{L}_{\text{ssl}}^{(k)}(\theta, \eta_k)\}_{k=1}^K$ and semi-supervised loss $\mathcal{L}_{\text{task}}(\theta, \omega)$ are *Joint Training* (`JT`) and *Pre-train&Fine-tune* (`P&F`), as shown in Fig. 2. The *Joint Training* strategy jointly trains the entire model under the supervision of downstream and pretext tasks, which can be considered as a kind of multi-task learning, defined as

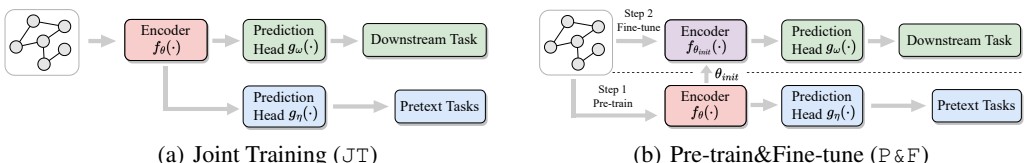

(a) Joint Training (JT)         (b) Pre-train&Fine-tune (P&F)

Figure 2: Illustration of the two training strategies, namely Joint Training and Pre-train&Fine-tune.

$$\min_{\theta,\omega,\{\eta_k\}_{k=1}^K} \mathcal{L}_{\text{task}}(\theta,\omega) + \alpha \sum_{k=1}^{K} \lambda_k \mathcal{L}_{\text{ssl}}^{(k)}(\theta,\eta_k), \tag{1}$$

where $\alpha$ is a trade-off hyperparameter and $\{\lambda_k\}_{k=1}^K$ are task weights. The *Pre-train&Fine-tune* strategy works in a two-stage manner: (1) Pre-training the GNN encoder $f_\theta(\cdot)$ with self-supervised pretext tasks; and (2) Fine-tuning the pre-trained GNN encoder $f_{\theta_{init}}(\cdot)$ with a prediction head $g_\omega(\cdot)$ under the supervision of a specific downstream task. The learning objective can be formulated as

$$\min_{(\theta,\omega)} \mathcal{L}_{\text{task}}(\theta_{init},\omega), \text{s.t. } \theta_{init}, \{\eta_k^*\}_{k=1}^K = \arg\min_{\theta,\{\eta_k\}_{k=1}^K} \sum_{k=1}^{K} \lambda_k \mathcal{L}_{\text{ssl}}^{(k)}(\theta,\eta_k). \tag{2}$$

A high-level overview of the two strategies is shown in Fig. 2. Without loss of generality, we mainly introduce our model for the P&F strategy, leaving extensions to the JT strategy in **Appendix A.1**.

A vanilla solution to combine multiple pretext tasks is to set the task weight $\lambda_k = \frac{1}{K}$ $(1 \le k \le K)$, i.e., to treat different tasks as equally important, but this completely ignores the importance of different tasks. Different from hand-crafted task weights, AutoSSL (Jin et al., 2021) and ParetoGNN (Ju et al., 2022) propose to learn a set of task weights $\{\lambda_k\}_{k=1}^K$ by some predefined priors (e.g., graph homogeneity or Pareto optimality), such that $f_\theta(\cdot)$ trained with the weighted loss $\sum_{k=1}^{K} \lambda_k \mathcal{L}_{\text{ssl}}^{(k)}(\theta,\eta_k)$ can extract meaningful representations. Despite the great progress, they only globally learn a dataset-specific loss weight for each task, while completely ignoring the *instance-level* requirement that different instances (nodes) may have localized task preferences. In practice, it is difficult to extend loss weighting directly from the task level to the instance level; for example, the loss function of PAIRDIS involves two nodes, which is hardly compatible with the node-specific loss function of PAR. Therefore, we would like to develop an instance-level multi-task SSL framework that captures the knowledge behind each pretext task by training multiple teachers, and then integrates the knowledge of different teachers separately for each instance in the fine-tuning phase, instead of global loss weighting in the pre-training phase. More importantly, compared to loss weighting during pre-training, knowledge integration in the fine-tuning phase can fully utilize downstream supervision to learn not only dataset-specific but also task-specific SSL strategies.

## 3 METHODOLOGY

### 3.1 MULTI-TEACHER KNOWLEDGE DISTILLATION

Intuitively, training with multiple pretext tasks enables the model to access richer information, which is beneficial for improving performance. However, this holds true only if we can well handle the compatibility problem between pretext tasks. To this end, we propose in this paper a novel multi-teacher knowledge distillation framework, as shown in Fig. 3, where we train multiple teachers with different pretext tasks to extract different levels of knowledge, which are then integrated through an instance-level knowledge integration module $\lambda_\gamma(\cdot,\cdot)$ and finally distilled into the student model. In the pre-training phase, we pre-trained each teacher model with a different pretext task, as follows

$$\theta_k^{init}, \eta_k^* = \arg\min_{\theta_k,\eta_k} \mathcal{L}_{\text{ssl}}^{(k)}(\theta_k,\eta_k), \text{ where } 1 \le k \le K. \tag{3}$$

In the fine-tuning phase, we fine-tune each teacher $\{\theta_k^{init},\omega_k\}$ with downstream supervision, then integrate the knowledge of different teachers, and distill it into the student $\{\theta,\omega\}$, as follows

$$\min_{\theta,\omega,\gamma} \mathcal{L}_{\text{task}}(\theta,\omega) + \beta \frac{\tau^2}{N} \sum_{i=1}^{N} \mathcal{L}_{KL}\left(\widetilde{\mathbf{z}}_i, \sum_{k=1}^{K} \lambda_\gamma(k,i)\widetilde{\mathbf{h}}_i^{(k)}\right), \text{s.t. } \theta_k^*,\omega_k^* = \arg\min_{(\theta_k,\omega_k)} \mathcal{L}_{\text{task}}(\theta_k^{init},\omega_k) \tag{4}$$

where $\mathcal{L}_{KL}(\cdot,\cdot)$ is the KL-divergence loss, $\beta$ is a trade-off hyperparameter, $\tau$ is the distillation temperature, and $\tau^2$ is used to keep the gradient stability (Hinton et al., 2015). In addition, $\widetilde{\mathbf{z}}_i =$

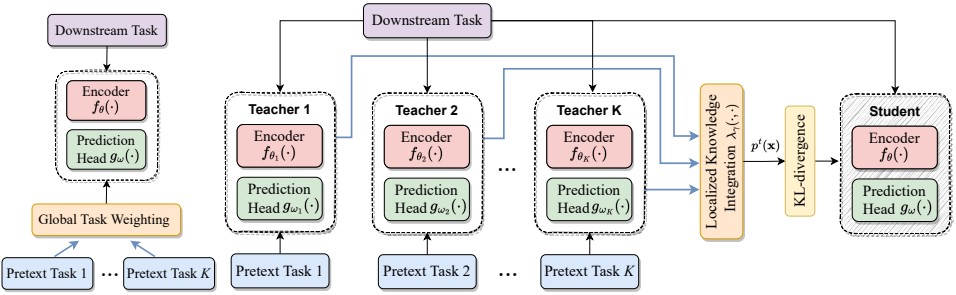

(a) Joint Training  (b) Multi-teacher Knowledge Distillation for Multi-tasking Graph SSL

Figure 3: **(a)** Conventional multi-tasking self-supervised learning where the model is jointly trained with multiple (globally) weighted pretext tasks. **(b)** Proposed multi-teacher knowledge distillation framework, where we train each teacher with one pretext task, and then apply an instance-level integration module to integrate the knowledge of different teachers for each instance separately.

$\sigma(\mathbf{z}_i/\tau)$, $\widetilde{\mathbf{h}}_i^{(k)} = \sigma(\mathbf{h}_i^{(k)}/\tau)$, $\sigma(\cdot) = \text{softmax}(\cdot)$ is the activation function, and $\mathbf{z}_i = g_\omega(f_\theta(\mathcal{G}, i))$ and $\mathbf{h}_i^{(k)} = g_{\omega_k^*}(f_{\theta_k^*}(\mathcal{G}, i))$ are the logits of node $v_i$ in the student model and $k$-th teacher model, respectively. MGSSL takes full account of the instance-level requirements and learns a customized knowledge integration strategy for each instance by a parameterized function $\lambda_\gamma(\cdot, \cdot)$, where $\lambda_\gamma(k, i)$ denotes the importance weight of $k$-th pretext task for node $v_i$, and it satisfies $\sum_{k=1}^{K} \lambda_\gamma(k, i) = 1$. The parameters to be optimized in Eq. (4) during KD are the student model $\{\theta, \omega\}$ and the weighting function $\lambda_\gamma(\cdot, \cdot)$ (parameterized by $\gamma$). Although each teacher model is frozen before KD, the integrated teacher (the optimality of teacher) $\sum_{k=1}^{K} \lambda_\gamma(k, i)\widetilde{\mathbf{h}}_i^{(k)}$ changes as $\lambda_\gamma(k, i)$ is updated during KD. Therefore, Eq. (4) essentially performs multi-teacher KD in an online fashion.

### 3.2 TWO PARAMETERIZED KNOWLEDGE INTEGRATION SCHEMES

A natural solution to achieve instance-level knowledge integration is to introduce a weighting function $\lambda_\gamma(\cdot \mid \gamma_i)$ parameterized by $\gamma_i \in \mathbb{R}^F$. However, directly fitting each $\lambda_\gamma(\cdot \mid \gamma_i)$ $(1 \le i \le N)$ locally involves solving $NF$ parameters, which increases the over-fitting risk, given the limited labels in the graph. Therefore, we consider the amortization inference (Kingma & Welling, 2013) which avoids the optimization of parameter $\gamma_i$ for each node locally and instead fits a shared neural network. In this section, we introduce two knowledge integration schemes, MGSSL-LF and MGSSL-TS, to parameterize the weighting function $\lambda_\gamma(\cdot, \cdot)$, resulting in two specific instantiations.

**MGSSL-LF**. To explicitly capture the localized importance of different teachers, we introduce a set of latent variables $\{\boldsymbol{\mu}_k\}_{k=1}^K$ and associate each teacher with a latent factor $\boldsymbol{\mu}_k \in \mathbb{R}^C$ to represent it. This scheme is inspired by latent factor models commonly applied in the recommender system (Koren, 2008), where each user or item corresponds to one latent factor used to summarize their implicit features. The importance weight of the $k$-th teacher to node $v_i$ can be calculated as follows

$$\lambda_\gamma(k, i) = \frac{\exp\left(\zeta_{k,i}\right)}{\sum_{k'=1}^{K} \exp\left(\zeta_{k',i}\right)}, \quad \text{where} \quad \zeta_{k,i} = \boldsymbol{\nu}^T\left(\boldsymbol{\mu}_k \odot \mathbf{z}_i\right). \quad (5)$$

where $\boldsymbol{\nu} \in \mathbb{R}^C$ is a global parameter vector to be learned, which determines whether or not the value of each dimension in $\left(\boldsymbol{\mu}_k \odot \mathbf{z}_i\right)$ has a positive effect on the importance score. Larger $\lambda_\gamma(k, i)$ denotes that the knowledge extracted by $k$-th teacher is more important to node $v_i$.

**MGSSL-TS**. Unlike MGSSL-LF, which calculates importance weights based solely on the node embeddings of different teachers, MGSSL-TS takes into account the matching degree of each teacher-student pair to distill the most matched teacher knowledge into the student model. We separately project the node logits of the student $\mathbf{z}_i = g_\omega(f_\theta(\mathcal{G}, i)) \in \mathbb{R}^C$ and each teacher $\mathbf{h}_i^{(k)} = g_{\omega_k^*}(f_{\theta_k^*}(\mathcal{G}, i)) \in \mathbb{R}^C$ into two subspaces via a linear transformation $\mathbf{W} \in \mathbb{R}^{C \times C}$. Then, the importance weight of $k$-th teacher (e.g., pretext task) to node $v_i$ can be calculated as follows

$$\lambda_\gamma(k, i) = \frac{\exp\left(\zeta_{k,i}\right)}{\sum_{k'=1}^{K} \exp\left(\zeta_{k',i}\right)}, \quad \text{where} \quad \zeta_{k,i} = \left(\mathbf{W}\mathbf{z}_i\right)^T\left(\mathbf{W}\mathbf{h}_i^{(k)}\right). \quad (6)$$

### 3.3 THEORETICAL GUIDELINE FOR HOW TO INTEGRATE

We have established a unified MGSSL framework in Sec. 3.1 and designed two schemes to parameterize $\lambda_\gamma(\cdot,\cdot)$ in Sec. 3.2. One more problem left to be solved is what is the criterion for knowledge integration, that is, how to optimize the learning of $\lambda_\gamma(\cdot,\cdot)$. In this section, we **(P1)** establish a provable theoretical guideline that tells us how to integrate, i.e., *what is the criteria for constructing a relatively "good" integrated teacher*; **(P2)** provide a theory-guided practical implementation; and **(P3)** present a theoretical justification for the potential of MGSSL to benefit from more teachers.

#### 3.3.1 PROVABLE THEORETICAL GUIDELINE

Let's define $R(\theta,\omega) = \mathbb{E}_\mathbf{x}\big[\mathbb{E}_{y|\mathbf{x}}\big[\ell\big(y, \sigma(g_\omega(f_\theta(\mathbf{x}))/\tau)\big)\big]\big] = \mathbb{E}_\mathbf{x}\big[\mathbf{p}^*(\mathbf{x})^\top \mathbf{1}\big(g_\omega(f_\theta(\mathbf{x}))\big)\big]$ as ***Bayesian objective***, where $\mathbf{p}^*(x) \doteq [\mathbb{P}(y|x)]_{y\in[C]}$ is the Bayesian class-probability. Besides, $\mathbf{1}\big(g_\omega(f_\theta(\mathbf{x}))\big) = \big(\ell(1, \sigma(g_\omega(f_\theta(\mathbf{x}))/\tau)), \cdots, \ell(C, \sigma(g_\omega(f_\theta(\mathbf{x}))/\tau))\big)$ is the loss vector, where $\ell(\cdot,\cdot)$ is the cross-entropy loss and $C$ is the number of category. We set $\mathbf{p}^\mathrm{t}(\mathbf{x}_i) \doteq \sum_{k=1}^K \lambda_\gamma(k,i)\widetilde{\mathbf{h}}_i^{(k)}$ to simplify the notations and rewrite the distillation term of Eq. (4) as a ***distillation objective***, as follows

$$\frac{1}{N}\sum_{i=1}^N \mathcal{L}_{KL}\Big(\widetilde{\mathbf{z}}_i, \sum_{k=1}^K \lambda_\gamma(k,i)\widetilde{\mathbf{h}}_i^{(k)}\Big) \propto \frac{1}{N}\sum_{i=1}^N \mathbf{p}^\mathrm{t}(\mathbf{x}_i)^\top \mathbf{1}\big(g_\omega(f_\theta(\mathbf{x}_i))\big) \doteq \widetilde{R}(\theta,\omega), \qquad (7)$$

where the detailed derivation of Eq. (7) is available in **Appendix A.2**. Previous work (Menon et al., 2021) has provided a statistical perspective on single-teacher knowledge distillation, where a Bayesian teacher providing true class probabilities $\{\mathbf{p}^*(\mathbf{x}_i)\}_{i=1}^N$ can lower the variance of the ***downstream objective*** $\mathcal{L}_{\mathrm{task}}(\theta,\omega) = \frac{1}{N}\sum_{i=1}^N \mathbf{e}_{y_i}^\top \mathbf{1}\big(g_\omega(f_\theta(\mathbf{x}_i))\big)$, where $\mathbf{e}_{y_i}^\top$ is the one-hot label of node $v_i$; the reward of reducing variance is beneficial for improving generalization (Maurer & Pontil, 2009). However, the teacher probabilities $\{\widetilde{\mathbf{h}}_i^{(k)}\}_{k=1}^K$ and Bayesian probability $\mathbf{p}^*(\mathbf{x}_i)$ are very likely to be *linearly independent* in the multi-teacher distillation framework, which means that we cannot guarantee $\mathbf{p}^\mathrm{t}(\mathbf{x}_i) = \sum_{k=1}^K \lambda_\gamma(k,i)\widetilde{\mathbf{h}}_i^{(k)} = \mathbf{p}^*(\mathbf{x}_i)$ for node $v_i \in \mathcal{V}$ by just adjusting weights $\{\lambda_\gamma(k,i)\}_{k=1}^K$. In practice, the following Proposition 1 indicates that even an imperfect teacher $\mathbf{p}^\mathrm{t}(\mathbf{x}) \neq \mathbf{p}^*(\mathbf{x})$ can still improve model generalization by approximating the Bayesian teacher $\mathbf{p}^*(\mathbf{x})$.

**Proposition 1** *Consider a Bayesian teacher $\mathbf{p}^*(\mathbf{x})$ and an integrated teacher $\mathbf{p}^\mathrm{t}(\mathbf{x})$. Given $N$ training samples $S = \{\mathbf{x}_i\}_{i=1}^N \sim \mathbb{P}^N$, the difference between the distillation objective $\widetilde{R}(\theta,\omega)$ and Bayesian objective $R(\theta,\omega)$ is bounded by Mean Square Errors (MSE) of their probabilities,*

$$\mathbb{E}_{S\sim\mathbb{P}^N}\Big[\big(\widetilde{R}(\theta,\omega) - R(\theta,\omega)\big)^2\Big] \leq \frac{1}{N}\mathbb{V}_{S\sim\mathbb{P}^N}\Big[\mathbf{p}^\mathrm{t}(\mathbf{x})^\top \mathbf{1}\big(g_\omega(f_\theta(\mathbf{x}))\big)\Big] + \mathcal{O}\Big(\mathbb{E}_\mathbf{x}\big[\|\mathbf{p}^\mathrm{t}((\mathbf{x})) - \mathbf{p}^*((\mathbf{x}))\|_2\big]\Big)^2 \quad (8)$$

where $\mathbb{P}$ is the data distribution of input data $\mathbf{x}$, and the derivation of Eq. (8) is available in **Appendix A.3**. On the right-hand side of Eq. (8), the second term $\mathcal{O}\Big(\mathbb{E}_\mathbf{x}\big[\|\mathbf{p}^\mathrm{t}(x) - \mathbf{p}^*(x)\|_2\big]\Big)^2$ dominates when $N$ is sufficiently large, which suggests that the effectiveness of knowledge distillation is governed by how close the teacher probability $\mathbf{p}^\mathrm{t}(\mathbf{x})$ are to the Bayesian probability $\mathbf{p}^*(\mathbf{x})$. The above discussion reached a theoretical guidance 1 for how to optimize $\lambda_\gamma(\cdot,\cdot)$ for knowledge integration.

**Guidance 1** *The instance-level knowledge weights should be set (or learned) in such a way that the integrated teacher probability $\mathbf{p}^\mathrm{t}(\mathbf{x})$ is as close as possible to the true Bayesian probability $\mathbf{p}^*(\mathbf{x})$.*

Two heuristic schemes for integrating different levels of knowledge from multiple teachers are averaged and label-based weighted integration. However, the averaged and weighted schemes have little to do with Guidance 1, and they are at potential risk of failing to differentiate important teachers from irrelevant ones and misleading the student in the presence of low-quality teachers. An intuitive illustration of this problem is provided in Fig. 4, where the integrated teacher probability $\mathbf{p}^\mathrm{t}(\mathbf{x})$ obtained by the averaged and weighted schemes not only does not come close but even deviates from

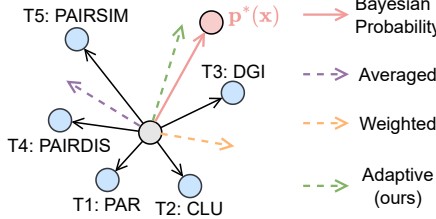

Figure 4: Illustration of the (2D) teacher probability directions for three schemes.

the true Bayesian probability $\mathbf{p}^*(\mathbf{x})$. Compared to heuristic schemes, this paper proposes two parameterized knowledge integration schemes that adaptively adjust the knowledge weights to meet Guideline 1, which enables the integrated teacher probability closer to the true Bayesian probability.

### 3.3.2 THEORY-GUIDED IMPLEMENTATION

In practice, precisely estimating the squared error to $\mathbf{p}^*(\mathbf{x})$ by Guidance 1 is not feasible (since $\mathbf{p}^*(\mathbf{x})$ is usually unknown), but one can estimate the quality of the teacher probability $\mathbf{p}^{\mathrm{t}}(\mathbf{x})$ on a holdout set, e.g., by computing the log-loss or squared loss over one-hot labels. This inspired us to approximately treat $\mathbf{p}^*(\mathbf{x}) \approx \mathbf{e}_y$ on the training set and optimize $\lambda_\gamma(\cdot, \cdot)$ by minimizing the cross-entropy loss $\mathcal{L}_W = \frac{1}{|\mathcal{V}_L|} \sum_{i \in \mathcal{V}_L} \ell(\mathbf{p}^{\mathrm{t}}(\mathbf{x}), \mathbf{p}^*(\mathbf{x}))$. The learned $\lambda_\gamma(\cdot, \cdot)$ can then be used to infer the proper teacher probability $\mathbf{p}^{\mathrm{t}}(\mathbf{x}_i)$ for unlabeled data $v_i \in \mathcal{V}_U$. While such estimations are often imperfect, they help to detect poor teacher probabilities, especially for those unlabeled data. Such an approximate estimation method was originally proposed by Menon et al. (2021), where a large number of simulation experiments are provided to demonstrate the effectiveness of such estimation from a statistical perspective. In this paper, we extend it from single-teacher distillation to a multi-teacher distillation setting and take it as a criterion to guide the optimization of the weighting function $\lambda_\gamma(\cdot, \cdot)$. The mean squared errors over one-hot labels on the training and testing sets in Fig. 7 have demonstrated the effectiveness of such estimations when $\mathbf{p}^*(\mathbf{x})$ is unknown in practice.

### 3.3.3 THEORETICAL JUSTIFICATION

Next, we derive the following Theorem 1, a theoretical justification to demonstrate the advantages of MGSSL under the multi-task learning setting, which theoretically proves that the **optimal** integrated teacher $\mathbf{p}^{\mathrm{t}}(x)$ can monotonically approximate $\mathbf{p}^*(x)$ as the number of teachers $K$ increases.

**Theorem 1** *Define* $\Delta(K) = \min \left\| \mathbf{p}^{\mathrm{t}}(\mathbf{x}_i) - \mathbf{p}^*(\mathbf{x}_i) \right\|_2 = \min \left\| \sum_{k=1}^{K} \lambda_\gamma(k, i) \widetilde{\mathbf{h}}_i^{(k)} - \mathbf{p}^*(\mathbf{x}_i) \right\|_2$ *with* $K(K \geq 1)$ *given teachers, then we have (1)* $\Delta(K + 1) \leq \Delta(K)$*, and (2)* $\lim_{K \to \infty} \Delta(K) = 0$*.*

where the above derivation is available in **Appendix A.4**. The theorem 1 indicates MGSSL is endowed with the theoretical potential to benefit from more teachers, i.e., it has advantages in handling the task-level compatibility, which is also supported by the experimental results in Sec. 4.3. The pseudo-code of the proposed MGSSL framework is summarized in Algorithm 1 in **Appendix A.5**.

## 4 EXPERIMENTAL EVALUATION

In this section, we evaluate MGSSL on eight datasets by answering five questions. **Q1**: Can MGSSL achieve better performance compared to training with individual tasks? **Q2:** How does MGSSL compare to state-of-the-art graph SSL baselines? **Q3:** Can MGSSL learn instance-level and customized SSL task combinations? **Q4:** Can MGSSL learn high-quality integrated teacher probabilities $\mathbf{p}^{\mathrm{t}}(\mathbf{x})$? **Q5:** How do the performance of MGSSL-LF and MGSSL-TS compare to other heuristics knowledge integration approaches? Can MGSSL consistently benefit from multiple teachers (tasks)?

**Dataset.** The effectiveness of the MGSSL framework is evaluated on eight real-world datasets, including Cora (Sen et al., 2008), Citeseer (Giles et al., 1998), Pubmed (McCallum et al., 2000), Coauthor-CS, Coauthor-Physics, Amazon-Photo, Amazon-Computers (Shchur et al., 2018), and ogbn-arxiv (Hu et al., 2020). A statistical overview of these eight datasets is placed in **Appendix A.6**. Each set of experiments is run five times with different random seeds, and the average accuracy and standard deviation are reported as performance metrics. Due to space limitations, we defer the implementation details and the best hyperparameter settings for each dataset to **Appendix A.7**.

**Baseline.** To evaluate the capability of MGSSL in multi-tasking graph SSL, we follow Jin et al. (2021) to consider five classical tasks (1) PAR (You et al., 2020b), which predicts pseudo-labels from graph partitioning; (2) CLU (You et al., 2020b), which predicts pseudo-labels from $K$-means clustering on node features; (3) DGI (Velickovic et al., 2019), which maximizes the mutual information between graph and node representations; (4) PAIRDIS (Jin et al., 2020), which predicts the shortest path length between nodes; and (5) PAIRSIM (Jin et al., 2020), which predicts the feature similarity between nodes. The detailed methodologies for these five tasks and the reasons why we selected them can be found in **Appendix A.8**. Moreover, we compare MGSSL with some representative SSL baselines in Table. 2, including GMI (Peng et al., 2020), MVGRL (Hassani & Khasahmadi,

Table 1: Performance comparison of single- and multi-task learning, where **bold** and underline denote the best metrics in multi- and single-task learning. Besides, we mark those metrics in multi-task learning that are poorer to vanilla GCNs and (the best) single-task learning as red and blue.

| Dataset | Setting | GCNs | Single Self-Supervised Task Learning | | | | | Multi Self-Supervised Task Learning | | | | |
|---|---|---|---|---|---|---|---|---|---|---|---|---|
| | | | PAR | CLU | DGI | PAIRDIS | PAIRSIM | Vanilla | AutoSSL | ParetoGNN | MGSSL-LF | MGSSL-TS |
| Cora | JT | $81.72_{\pm0.52}$ | $83.52_{\pm0.39}$ | $82.34_{\pm0.46}$ | $83.28_{\pm0.33}$ | $82.92_{\pm0.41}$ | $83.16_{\pm0.38}$ | $81.50_{\pm0.40}$ | $83.78_{\pm0.45}$ | $83.56_{\pm0.41}$ | $84.68_{\pm0.39}$ | $\mathbf{85.32}_{\pm0.32}$ |
| | P&F | $81.72_{\pm0.52}$ | $82.38_{\pm0.31}$ | $81.42_{\pm0.35}$ | $82.10_{\pm0.44}$ | $81.92_{\pm0.42}$ | $82.44_{\pm0.36}$ | $80.74_{\pm0.38}$ | $82.96_{\pm0.43}$ | $83.34_{\pm0.41}$ | $84.22_{\pm0.28}$ | $\mathbf{84.38}_{\pm0.27}$ |
| Citeseer | JT | $71.48_{\pm0.46}$ | $72.72_{\pm0.36}$ | $72.14_{\pm0.50}$ | $73.08_{\pm0.45}$ | $73.16_{\pm0.42}$ | $72.90_{\pm0.45}$ | $72.30_{\pm0.50}$ | $73.30_{\pm0.37}$ | $73.54_{\pm0.45}$ | $\mathbf{74.34}_{\pm0.31}$ | $74.20_{\pm0.40}$ |
| | P&F | $71.48_{\pm0.46}$ | $72.36_{\pm0.58}$ | $71.84_{\pm0.49}$ | $72.52_{\pm0.37}$ | $72.22_{\pm0.53}$ | $71.98_{\pm0.62}$ | $71.64_{\pm0.49}$ | $72.76_{\pm0.44}$ | $72.98_{\pm0.51}$ | $73.58_{\pm0.56}$ | $\mathbf{73.70}_{\pm0.76}$ |
| Pubmed | JT | $79.26_{\pm0.40}$ | $82.16_{\pm0.54}$ | $80.92_{\pm0.36}$ | $81.50_{\pm0.43}$ | $81.22_{\pm0.55}$ | $80.50_{\pm0.54}$ | $80.86_{\pm0.50}$ | $82.72_{\pm0.35}$ | $\mathbf{82.90}_{\pm0.40}$ | $82.66_{\pm0.32}$ | $82.82_{\pm0.29}$ |
| | P&F | $79.26_{\pm0.40}$ | $79.56_{\pm0.39}$ | $79.12_{\pm0.47}$ | $79.90_{\pm0.52}$ | $79.64_{\pm0.48}$ | $79.34_{\pm0.60}$ | $78.90_{\pm0.54}$ | $80.14_{\pm0.41}$ | $79.95_{\pm0.47}$ | $\mathbf{80.62}_{\pm0.25}$ | $80.54_{\pm0.42}$ |
| CS | JT | $91.04_{\pm0.45}$ | $92.30_{\pm0.67}$ | $92.94_{\pm0.70}$ | $92.66_{\pm0.69}$ | $92.48_{\pm0.55}$ | $93.12_{\pm0.64}$ | $92.16_{\pm0.60}$ | $93.54_{\pm0.46}$ | $93.38_{\pm0.42}$ | $\mathbf{93.86}_{\pm0.36}$ | $93.46_{\pm0.25}$ |
| | P&F | $91.04_{\pm0.45}$ | $91.28_{\pm0.55}$ | $91.36_{\pm0.63}$ | $91.80_{\pm0.73}$ | $91.44_{\pm0.49}$ | $91.62_{\pm0.47}$ | $91.42_{\pm0.57}$ | $\mathbf{92.48}_{\pm0.45}$ | $92.24_{\pm0.49}$ | $92.36_{\pm0.45}$ | $91.94_{\pm0.33}$ |
| Physics | JT | $93.06_{\pm0.55}$ | $94.08_{\pm0.56}$ | $94.12_{\pm0.49}$ | $94.74_{\pm0.46}$ | $94.62_{\pm0.63}$ | $94.40_{\pm0.48}$ | $93.94_{\pm0.47}$ | $95.10_{\pm0.42}$ | $95.28_{\pm0.48}$ | $\mathbf{95.74}_{\pm0.38}$ | $95.54_{\pm0.35}$ |
| | P&F | $93.06_{\pm0.55}$ | $93.18_{\pm0.71}$ | $93.50_{\pm0.53}$ | $93.92_{\pm0.60}$ | $94.04_{\pm0.56}$ | $93.34_{\pm0.73}$ | $93.40_{\pm0.50}$ | $93.88_{\pm0.45}$ | $93.43_{\pm0.57}$ | $94.80_{\pm0.29}$ | $\mathbf{94.96}_{\pm0.43}$ |
| Photo | JT | $91.90_{\pm0.46}$ | $92.54_{\pm0.60}$ | $93.04_{\pm0.55}$ | $92.46_{\pm0.70}$ | $92.32_{\pm0.55}$ | $92.82_{\pm0.78}$ | $91.52_{\pm0.61}$ | $92.94_{\pm0.40}$ | $92.76_{\pm0.50}$ | $93.98_{\pm0.29}$ | $\mathbf{94.22}_{\pm0.31}$ |
| | P&F | $91.90_{\pm0.46}$ | $92.24_{\pm0.49}$ | $92.58_{\pm0.66}$ | $92.02_{\pm0.59}$ | $92.10_{\pm0.52}$ | $92.42_{\pm0.44}$ | $90.84_{\pm0.51}$ | $92.36_{\pm0.49}$ | $92.78_{\pm0.54}$ | $93.32_{\pm0.37}$ | $\mathbf{93.52}_{\pm0.41}$ |
| Computers | JT | $86.36_{\pm0.65}$ | $87.48_{\pm0.65}$ | $87.96_{\pm0.72}$ | $88.08_{\pm0.64}$ | $87.62_{\pm0.52}$ | $88.40_{\pm0.72}$ | $86.58_{\pm0.50}$ | $88.72_{\pm0.44}$ | $88.90_{\pm0.47}$ | $89.56_{\pm0.34}$ | $\mathbf{89.72}_{\pm0.28}$ |
| | P&F | $86.36_{\pm0.65}$ | $86.72_{\pm0.78}$ | $87.74_{\pm0.80}$ | $87.36_{\pm0.73}$ | $86.52_{\pm0.65}$ | $87.20_{\pm0.69}$ | $85.90_{\pm0.57}$ | $88.00_{\pm0.49}$ | $88.14_{\pm0.63}$ | $\mathbf{88.68}_{\pm0.42}$ | $88.42_{\pm0.33}$ |
| ogbn-arxiv | JT | $71.16_{\pm0.32}$ | $71.84_{\pm0.28}$ | $71.72_{\pm0.40}$ | $72.04_{\pm0.25}$ | $72.18_{\pm0.30}$ | $71.90_{\pm0.33}$ | $70.94_{\pm0.33}$ | $72.26_{\pm0.25}$ | $72.30_{\pm0.23}$ | $72.66_{\pm0.26}$ | $\mathbf{72.72}_{\pm0.22}$ |
| | P&F | $71.16_{\pm0.32}$ | $71.78_{\pm0.37}$ | $71.54_{\pm0.36}$ | $71.96_{\pm0.28}$ | $71.90_{\pm0.33}$ | $71.62_{\pm0.29}$ | $70.56_{\pm0.31}$ | $72.08_{\pm0.24}$ | $72.24_{\pm0.27}$ | $72.52_{\pm0.31}$ | $\mathbf{72.60}_{\pm0.25}$ |

Table 2: Performance comparison with classical self-supervised algorithms under the *Joint Training* setting, where **bold** and underline denote the best and second metrics on each dataset, respectively.

| Method | Cora | Citeseer | Pubmed | CS | Physics | Photo | Computers | ogbn-arxiv | Avg. Rank ↓ |
|---|---|---|---|---|---|---|---|---|---|
| GCNs | $81.72_{\pm0.52}$ | $71.48_{\pm0.46}$ | $79.26_{\pm0.40}$ | $91.04_{\pm0.45}$ | $93.06_{\pm0.55}$ | $91.90_{\pm0.46}$ | $86.36_{\pm0.65}$ | $71.16_{\pm0.32}$ | 12.13 |
| DGI | $83.28_{\pm0.33}$ | $73.08_{\pm0.45}$ | $81.50_{\pm0.43}$ | $92.66_{\pm0.69}$ | $94.74_{\pm0.46}$ | $92.46_{\pm0.70}$ | $88.08_{\pm0.64}$ | $72.04_{\pm0.25}$ | 9.63 |
| GMI | $82.94_{\pm0.40}$ | $73.22_{\pm0.38}$ | $81.20_{\pm0.35}$ | $92.76_{\pm0.56}$ | *OOM* | $92.74_{\pm0.56}$ | $88.20_{\pm0.45}$ | *OOM* | 10.00 |
| MVGRL | $83.36_{\pm0.43}$ | $72.66_{\pm0.37}$ | $81.74_{\pm0.41}$ | $92.84_{\pm0.39}$ | *OOM* | $93.06_{\pm0.45}$ | $88.36_{\pm0.51}$ | *OOM* | 8.67 |
| GRACE | $80.80_{\pm0.38}$ | $72.24_{\pm0.44}$ | $79.96_{\pm0.46}$ | $91.94_{\pm0.37}$ | $93.64_{\pm0.47}$ | $91.92_{\pm0.43}$ | $87.44_{\pm0.49}$ | *OOM* | 11.86 |
| GCA | $84.34_{\pm0.45}$ | $73.72_{\pm0.37}$ | $81.98_{\pm0.42}$ | $93.30_{\pm0.42}$ | $94.78_{\pm0.52}$ | $93.30_{\pm0.36}$ | $88.74_{\pm0.37}$ | *OOM* | 5.71 |
| GraphMAE | $84.20_{\pm0.40}$ | $73.40_{\pm0.40}$ | $81.10_{\pm0.40}$ | $93.44_{\pm0.41}$ | $94.56_{\pm0.48}$ | $93.54_{\pm0.45}$ | $88.90_{\pm0.43}$ | $71.75_{\pm0.17}$ | 6.75 |
| CG3 | $83.76_{\pm0.39}$ | $73.54_{\pm0.40}$ | $81.58_{\pm0.36}$ | $93.02_{\pm0.51}$ | $94.90_{\pm0.39}$ | $93.68_{\pm0.48}$ | $88.42_{\pm0.42}$ | $72.40_{\pm0.24}$ | 6.25 |
| BGRL | $84.82_{\pm0.41}$ | $73.96_{\pm0.35}$ | $82.20_{\pm0.34}$ | $93.58_{\pm0.29}$ | $95.12_{\pm0.44}$ | $93.48_{\pm0.51}$ | $89.08_{\pm0.38}$ | $\mathbf{72.80}_{\pm0.20}$ | 3.13 |
| AutoSSL | $83.78_{\pm0.45}$ | $73.30_{\pm0.57}$ | $82.72_{\pm0.35}$ | $93.54_{\pm0.46}$ | $95.10_{\pm0.42}$ | $92.94_{\pm0.40}$ | $88.72_{\pm0.44}$ | $72.26_{\pm0.25}$ | 5.75 |
| ParetoGNN | $83.56_{\pm0.41}$ | $73.54_{\pm0.45}$ | $\mathbf{82.90}_{\pm0.40}$ | $93.38_{\pm0.42}$ | $95.28_{\pm0.48}$ | $92.76_{\pm0.50}$ | $88.90_{\pm0.47}$ | $72.30_{\pm0.23}$ | 5.13 |
| MGSSL-LF | $84.68_{\pm0.39}$ | $\mathbf{74.34}_{\pm0.31}$ | $82.66_{\pm0.32}$ | $\mathbf{93.86}_{\pm0.36}$ | $\mathbf{95.74}_{\pm0.38}$ | $93.98_{\pm0.29}$ | $89.56_{\pm0.34}$ | $72.66_{\pm0.26}$ | 2.13 |
| MGSSL-TS | $\mathbf{85.32}_{\pm0.32}$ | $74.20_{\pm0.42}$ | $82.82_{\pm0.29}$ | $93.46_{\pm0.25}$ | $95.54_{\pm0.35}$ | $\mathbf{94.22}_{\pm0.31}$ | $\mathbf{89.72}_{\pm0.28}$ | $72.72_{\pm0.22}$ | 1.88 |

2020), GRACE (Zhu et al., 2020a), GCA (Zhu et al., 2020b), GraphMAE (Hou et al., 2022), CG3 (Wan et al., 2020), and BGRL (Thakoor et al., 2021), AutoSSL (Jin et al., 2021), and ParetoGNN (Ju et al., 2022). Due to space limitations, we defer the discussion of related work on graph SSL and automated learning to **Appendix A.9**. In this paper, we mainly demonstrate the effectiveness of MGSSL using the node classification task, but MGSSL also has the potential to be extended to other tasks, including graph regression (e.g, molecular property prediction), node clustering, link prediction, and vision tasks, and we place the relevant preliminary results in **Appendix A.10**.

## 4.1 PERFORMANCE COMPARISON

**Performance Comparison with Individual Tasks (Q1).** We report the results for single- and multi-tasking learning under two training strategies, i.e., *Joint Training* (JT) and *Pre-train&Fine-tune* (P&F) in Table. 1, from which we make three observations: (1) The performance of individual pretext tasks depends heavily on the datasets, and there does not exist an "optimal" task that works for all datasets. (2) Simply averaging task losses over all tasks (Vanilla) may cause a serious task-level compatibility problem, whose performance is not only inferior to training with individual tasks (marked in blue), but even poorer than vanilla GCNs (marked in red). (3) As an automated self-supervised learning approach, AutoSSL performs better than Vanilla, but still lags far behind our MGSSL overall on eight graph datasets. Apart from the results reported in Table. 1 with GCN (Kipf & Welling, 2016) as the backbone, we also experiment with GAT (Veličković et al., 2017) and GraphSAGE (Hamilton et al., 2017) as the backbones, respectively, in **Appendix A.11**.

**Performance Comparison with Representative SSL Baselines (Q2).** We compare MGSSL with several representative graph SSL baselines under the JT setting (the results under the P&F setting are placed in **Appendix A.12**). As can be seen from the results reported in Table. 2, by combining just a few simple and classical pretext tasks, the resulting performance is comparable to that of several state-of-the-art self-supervised baselines. For example, MGSSL-LF and MGSSL-TS perform better

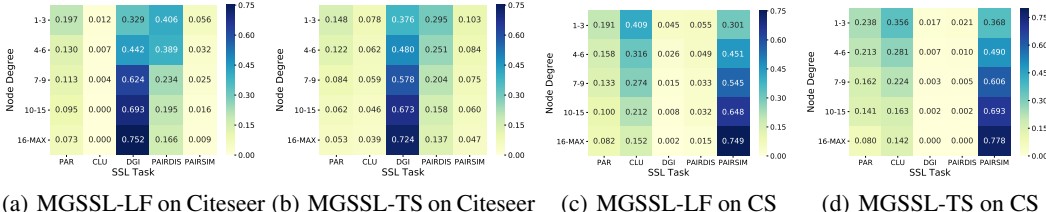

(a) MGSSL-LF on Citeseer (b) MGSSL-TS on Citeseer (c) MGSSL-LF on CS (d) MGSSL-TS on CS

Figure 5: Illustration of average knowledge weights for nodes with different node degree ranges.

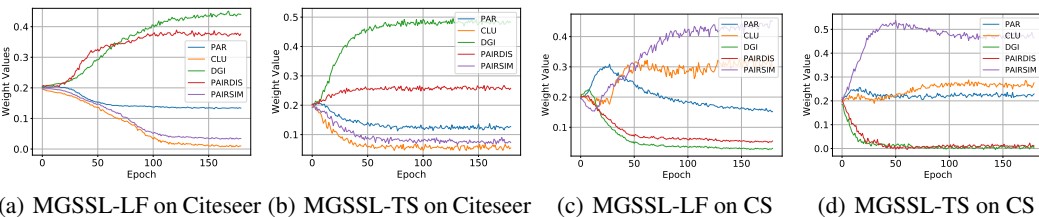

(a) MGSSL-LF on Citeseer (b) MGSSL-TS on Citeseer (c) MGSSL-LF on CS (d) MGSSL-TS on CS

Figure 6: Evolution process of average knowledge weights for nodes with a degree range of [4, 6].

than all other baselines on 5 out of 8 datasets. More importantly, we find that MGSSL outperforms previous multi-tasking SSL baselines, AutoSSL and ParetoGNN, by a large margin on eight datasets.

### 4.2 LOCALIZED SSL TASKS AND LEARNING CURVES

**Localized and Customized SSL Strategies (Q3).** To answer **Q3**, we visualize the average knowledge weights learned by MGSSL-LF and MGSSL-TS at different node degree ranges on the Citeseer and Coatuhor-CS datasets. From the heatmaps shown in Fig. 5, we can make three important observations: (1) The learned knowledge weights vary a lot from dataset to dataset. For example, Citeseer can benefit more from pretext tasks - DGI and PAIRDIS, while the tasks of CLU and PAIRSIM are more beneficial for Coauthor-CS. (2) The knowledge weights learned by MGSSL-LF and MGSSL-TS are very similar on the same dataset, suggesting that they do uncover some "essence". (3) The knowledge weights vary greatly across different node degrees, and this variation is almost monotonic. For example, as the node degree increases on Citeseer, the dependence of nodes on DGI increases, while the dependence on PAIRDIS gradually decreases, which indicates that MGSSL has advantages in learning instance-level and customized SSL strategies.

Furthermore, we also provide in Fig. 6 the evolution process of knowledge weights for nodes with a degree range of [4, 6] on the Citeseer and Coatuhor-CS datasets. The weights of five tasks eventually become stable and converge to steady values, corresponding to the results in Fig. 5. For instance, the weight of the CLU pretext task eventually converges to a value close to 0 in Fig. 6(a), at which point this task essentially quits training and contributes little to the performance improvement.

**Learning Curves (Q4).** Since the true Bayesian probability $\mathbf{p}^*(\mathbf{x})$ is often unknown in practice, it is not feasible to directly estimate the squared errors between $\mathbf{p}^t(\mathbf{x})$ and $\mathbf{p}^*(\mathbf{x})$. Therefore, we follow Menon et al. (2021) to estimate the quality of the teacher probability $\mathbf{p}^t(\mathbf{x})$ by computing the Mean Squared Errors (MSE) over one-hot labels. We provide the curves of MSE and accuracy during training in Fig. 7, from which we observe that the MSE gradually decreases while the accuracy gradually increases on both the training and testing sets as the training proceeds. This justifies the theoretical Guideline 1 and shows the effectiveness of the two knowledge integration schemes.

### 4.3 EVALUATION ON KNOWLEDGE INTEGRATION AND TEACHER NUMBER (Q5)

We compare MGSSL-LF and MGSSL-TS with three heuristic knowledge integration schemes, including (1) Random, setting $\lambda_\gamma(k, i)$ randomly in the range of [0,1]; (2) Average, setting $\lambda_\gamma(k, i) = 1/K$ throughout training, and (3) Weighted, calculating cross-entropy as weights on the labeled nodes, and using average weights for unlabeled nodes. For a fair comparison, we perform softmax activation for each scheme to satisfy $\sum_{k=1}^{K} \lambda_\gamma(k, i) = 1$. Note that all these schemes are implemented based on our multi-teacher KD framework. We provide the performance of these

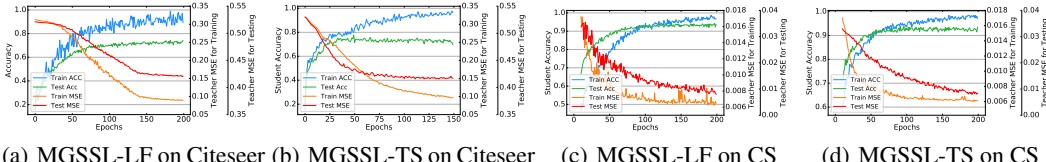

(a) MGSSL-LF on Citeseer (b) MGSSL-TS on Citeseer  (c) MGSSL-LF on CS  (d) MGSSL-TS on CS

Figure 7: Illustrations of the learning curves of *(a-b)* Mean Squared Errors (MSE) of teacher probability $\mathbf{p}^{\mathrm{t}}(\mathbf{x})$ over the one-hot labels on the training and testing sets and *(c-d)* classification accuracy on the training and testing sets, to estimate the quality of the teacher probability $\mathbf{p}^{\mathrm{t}}(\mathbf{x})$.

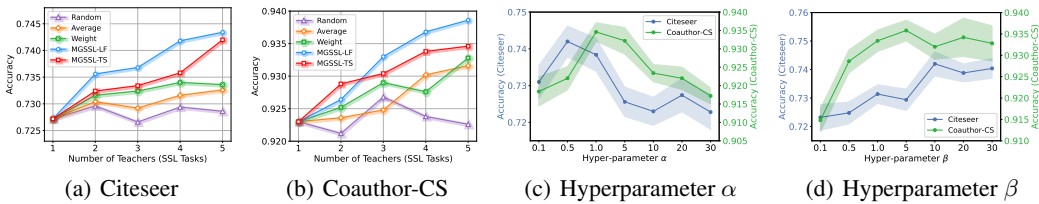

(a) Citeseer  (b) Coauthor-CS  (c) Hyperparameter $\alpha$  (d) Hyperparameter $\beta$

Figure 8: *(a-b)* Ablation study on knowledge integration under different numbers of teachers (with numerical values in **Appendix A.13**). *(c-d)* Parameter sensitivity analyses on loss weights $\alpha$ and $\beta$.

schemes under five different numbers of teachers in Fig. 8(a) and Fig. 8(b), from which we can make three observations: (1) Random does not benefit from multiple teachers and is even poorer than the one trained with one individual task; (2) Average and Weighted cannot always benefit from multiple teachers; for example, the Weighted scheme trained with five pretext task is inferior to the one trained with four pretext tasks on the Citeseer dataset; (3) MGSSL-LF and MGSSL-TS both perform better than the other three heuristics under various numbers of teachers. More importantly, both MGSSL-LF and MGSSL-TS can consistently benefit from more teachers, which aligns with Theorem 1. Further results on more teachers (up to 10 teachers) can be found in **Appendix A.14**.

## 4.4 PARAMETER SENSITIVITY & COMPUTATIONAL EFFICIENCY

We provide the hyperparameter sensitivity analysis on two key hyperparameters, e.g., loss weights $\alpha$ and $\beta$ in Fig. 8(c) and Fig. 8(d), from which it is clear that (1) setting the loss weight $\alpha$ of pretext tasks too large or too small is detrimental to extracting informative knowledge; (2) a large $\beta$ usually yields good performance, which illustrates the effectiveness of the distillation term in Eq. (4). In practice, we can determine $\alpha$ and $\beta$ by selecting the model with the highest accuracy on the validation set through the grid search. Due to space limitations, we place the analysis of the time complexity of MGSSL and the experimental results of the computational efficiency (i.e., the running time) in **Appendix A.15**, from which we find that compared to the joint training of multiple tasks by loss weighting, MGSSL not only does not increase but even has an advantage in the training time.

## 5 CONCLUSION

Over the past few years, there are hundreds of graph SSL algorithms proposed, which inspired us to move our attention away from designing more pretext tasks and towards making more effective use of those already on hand. In this paper, we propose a novel multi-teacher knowledge distillation framework for Multi-tasking Graph Self-Supervised Learning (MGSSL) to learn instance-level task preferences for each instance separately. More importantly, we provide a theoretical guideline and two adaptive knowledge integration schemes to integrate the knowledge from different teachers. Extensive experiments show that MGSSL can benefit from multiple pretext tasks and significantly improve the performance of individual tasks. While MGSSL automates the task selection for each node, it is still preliminary work, as how to construct a suitable pool of pretext tasks still requires human labor. In this sense, "full" automation is still desired and needs to be pursued in the future.

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

# APPENDIX

## A.1 EXTENSIONS TO THE *Joint Training*

To adapt `MGSSL` to the *Joint Training* setting, we defined the learning objective as follows

$$
\min_{\theta, \omega, \gamma} \mathcal{L}_{\text{task}}(\theta, \omega) + \beta \frac{\tau^2}{N} \sum_{i=1}^{N} \mathcal{L}_{KL}\Big((\widetilde{\mathbf{z}}_i), \sum_{k=1}^{K} \lambda_\gamma(k, i)(\widetilde{\mathbf{h}}_i^{(k)})\Big)
$$
$$
\text{s.t. } \theta_k^*, \omega_k^*, \eta_k^* = \underset{\theta_k, \omega_k, \eta_k}{\arg\min} \ \mathcal{L}_{\text{task}}(\theta_k, \omega_k) + \alpha \mathcal{L}_{\text{ssl}}^{(k)}(\theta_k, \eta_k), \text{ where } 1 \le k \le K
$$

(A.1)

## A.2 DISTILLATION OBJECTIVE REWRITING

We rewrite the distillation term of Eq. (4) in the form of $\widetilde{R}(\theta, \omega)$ in Eq. (7), as follows

$$
\frac{1}{N} \sum_{i=1}^{N} \mathcal{L}_{KL}\Big(\widetilde{\mathbf{z}}_i, \sum_{k=1}^{K} \lambda_\gamma(k, i)\widetilde{\mathbf{h}}_i^{(k)}\Big) = \frac{1}{N} \sum_{i=1}^{N} \mathcal{L}_{KL}\Big(\widetilde{\mathbf{z}}_i, \mathbf{p}^{\text{t}}(\mathbf{x}_i)\Big)
$$
$$
= \frac{1}{N} \sum_{i=1}^{N} \mathbf{p}^{\text{t}}(\mathbf{x}_i) \log \frac{\mathbf{p}^{\text{t}}(\mathbf{x}_i)}{\widetilde{\mathbf{z}}_i} = \frac{1}{N} \sum_{i=1}^{N} \mathcal{I}\big(\mathbf{p}^{\text{t}}(\mathbf{x}_i)\big) - \mathbf{p}^{\text{t}}(\mathbf{x}_i) \log \widetilde{\mathbf{z}}_i
$$

(A.2)

where $\mathcal{I}(\cdot)$ denotes the information entropy. In this paper, the distillation objective is used to mainly optimize parameters $f_\theta(\cdot)$ and $g_\omega(\cdot)$ of the student model and will not directly optimize the weighting function $\lambda_\gamma(k, i)$. As a result, although $\mathbf{p}^{\text{t}}(\mathbf{x}_i) = \sum_{k=1}^{K} \lambda_\gamma(k, i)\widetilde{\mathbf{h}}_i^{(k)}$ may be different from one training epoch to another, $\mathbf{p}^{\text{t}}(\mathbf{x})$ can be considered as *unoptimizable* in each training epoch. Therefore, we can directly omit the term $\mathcal{L}\big(\mathbf{p}^{\text{t}}(\mathbf{x}_i)\big)$ and derive the following proportional equation,

$$
\frac{1}{N} \sum_{i=1}^{N} \mathcal{L}\big(\mathbf{p}^{\text{t}}(\mathbf{x}_i)\big) - \mathbf{p}^{\text{t}}(\mathbf{x}_i) \log \widetilde{\mathbf{z}}_i \propto \frac{1}{N} \sum_{i=1}^{N} -\mathbf{p}^{\text{t}}(\mathbf{x}_i) \log \widetilde{\mathbf{z}}_i = \frac{1}{N} \sum_{i=1}^{N} \mathbf{p}^{\text{t}}(\mathbf{x}_i)^\top \mathbf{1}\big(g_\omega(f_\theta(\mathbf{x}_i))\big) \doteq \widetilde{R}(\theta, \omega)
$$

where $\mathbf{1}\big(g_\omega(f_\theta(\mathbf{x}_i))\big) = \big(\ell(1, \widetilde{\mathbf{z}}_i), \ell(2, \widetilde{\mathbf{z}}_i), \cdots, \ell(C, \widetilde{\mathbf{z}}_i)\big)$ denotes the cross-entropy loss vector.

## A.3 PROOF ON PROPOSITION 1

**Proposition 1** *Consider a Bayesian teacher $\mathbf{p}^*(\mathbf{x})$ and an integrated teacher $\mathbf{p}^{\text{t}}(\mathbf{x})$. Given $N$ training samples $S = \{\mathbf{x}_i\}_{i=1}^{N} \sim \mathbb{P}^N$, the difference between the distillation objective $\widetilde{R}(\theta, \omega)$ and Bayesian objective $R(\theta, \omega)$ is bounded by Mean Square Errors (MSE) of their probabilities,*

$$
\underset{S \sim \mathbb{P}^N}{\mathbb{E}} \left[ \big(\widetilde{R}(\theta, \omega) - R(\theta, \omega)\big)^2 \right] \le \frac{1}{N} \underset{S \sim \mathbb{P}^N}{\mathbb{V}} \Big[ \mathbf{p}^{\text{t}}(\mathbf{x})^\top \mathbf{1}\big(g_\omega(f_\theta(\mathbf{x}))\big) \Big] + \mathcal{O}\Big( \underset{\mathbf{x}}{\mathbb{E}} \big[\|\mathbf{p}^{\text{t}}((\mathbf{x})) - \mathbf{p}^*((\mathbf{x}))\|_2\big] \Big)^2 \quad \text{(A.3)}
$$

**Proof 1** *Given $N$ training samples $S = \{\mathbf{x}_i\}_{i=1}^{N} \sim \mathbb{P}^N$ randomly sampled from the data distribution $\mathbb{P}$ of input data $\mathbf{x}$, let's start the derivation from the left side of the equation, as follows*

$$
\underset{S \sim \mathbb{P}^N}{\mathbb{E}} \left[ \big(\widetilde{R}(\theta, \omega) - R(\theta, \omega)\big)^2 \right] = \underset{S \sim \mathbb{P}^N}{\mathbb{V}} \left[ \big(\widetilde{R}(\theta, \omega) - R(\theta, \omega)\big) \right] + \underset{S \sim \mathbb{P}^N}{\mathbb{E}} \left[ \big(\widetilde{R}(\theta, \omega) - R(\theta, \omega)\big) \right]^2 \quad \text{(A.4)}
$$

*Since $R(\theta, \omega) = \mathbb{E}_{\mathbf{x}} \big[\mathbf{p}^*(\mathbf{x})^\top \mathbf{1}\big(g_\omega(f_\theta(\mathbf{x}))\big)\big]$ will not change with the training samples $S$, we have*

$$
\underset{S \sim \mathbb{P}^N}{\mathbb{V}} \left[ \big(\widetilde{R}(\theta, \omega) - R(\theta, \omega)\big) \right] = \underset{S \sim \mathbb{P}^N}{\mathbb{V}} \left[ \widetilde{R}(\theta, \omega) \right] + \underset{S \sim \mathbb{P}^N}{\mathbb{V}} \left[ R(\theta, \omega) \right] - Cov\left( \widetilde{R}(\theta, \omega), R(\theta, \omega) \right)
$$
$$
= \underset{S \sim \mathbb{P}^N}{\mathbb{V}} \left[ \widetilde{R}(\theta, \omega) \right] = \frac{1}{N} \underset{S \sim \mathbb{P}^N}{\mathbb{V}} \left[ \mathbf{p}^{\text{t}}(\mathbf{x})^\top \mathbf{1}\big(g_\omega(f_\theta(\mathbf{x}))\big) \right]
$$

(A.5)

*Furthermore, we have*

$$
\underset{S \sim \mathbb{P}^N}{\mathbb{E}} R(\theta, \omega) = \underset{S \sim \mathbb{P}^N}{\mathbb{E}} \underset{\mathbf{x}}{\mathbb{E}} \big[\mathbf{p}^*(\mathbf{x})^\top \mathbf{1}\big(g_\omega(f_\theta(\mathbf{x}))\big)\big] = \underset{\mathbf{x}}{\mathbb{E}} \big[\mathbf{p}^*(\mathbf{x})^\top \mathbf{1}\big(g_\omega(f_\theta(\mathbf{x}))\big)\big] = R(\theta, \omega) \quad \text{(A.6)}
$$

503    *, and*

$$\mathop{\mathbb{E}}_{S\sim\mathbb{P}^N} \widetilde{R}(\theta,\omega) = \mathop{\mathbb{E}}_{S\sim\mathbb{P}^N} \frac{1}{N}\sum_{i=1}^{N} \mathbf{p}^{\mathrm{t}}(\mathbf{x}_i)^\top \mathbf{l}\big(g_\omega(f_\theta(\mathbf{x}_i))\big)$$

$$= \frac{1}{N}\mathop{\mathbb{E}}_{S\sim\mathbb{P}^N}\sum_{i=1}^{N} \mathbf{p}^{\mathrm{t}}(\mathbf{x}_i)^\top \mathbf{l}\big(g_\omega(f_\theta(\mathbf{x}_i))\big) \tag{A.7}$$

$$= \mathbb{E}_{\mathbf{x}}\left[\mathbf{p}^{\mathrm{t}}(\mathbf{x})^\top \mathbf{l}\big(g_\omega(f_\theta(\mathbf{x}_i))\big)\right]$$

504    *Therefore, we can derive the second term on the right-hand side in Eq. (A.4), as follows*

$$\mathop{\mathbb{E}}_{S\sim\mathbb{P}^N}\left[\left(\widetilde{R}(\theta,\omega) - R(\theta,\omega)\right)\right]^2 = \mathbb{E}_{\mathbf{x}}\left[\mathbf{p}^{\mathrm{t}}(\mathbf{x})^\top \mathbf{l}\big(g_\omega(f_\theta(\mathbf{x}))\big) - \mathbf{p}^*(\mathbf{x})^\top \mathbf{l}\big(g_\omega(f_\theta(\mathbf{x}))\big)\right]^2$$

$$= \mathbb{E}_{\mathbf{x}}\left[\big(\mathbf{p}^{\mathrm{t}}(\mathbf{x}) - \mathbf{p}^*(\mathbf{x})\big)^\top \mathbf{l}\big(g_\omega(f_\theta(\mathbf{x}))\big)\right]^2$$

$$\leq \mathbb{E}_{\mathbf{x}}\left[\left\|\mathbf{p}^{\mathrm{t}}(\mathbf{x}) - \mathbf{p}^*(\mathbf{x})\right\|_2 \cdot \left\|\mathbf{l}\big(g_\omega(f_\theta(\mathbf{x}))\big)\right\|_2\right]^2 \tag{A.8}$$

$$\doteq \mathcal{O}\left(\mathbb{E}_{\mathbf{x}}\left[\|\mathbf{p}^{\mathrm{t}}(x) - \mathbf{p}^*(x)\|_2\right]\right)^2$$

505    *where the inequality holds according to Cauchy-Schwartz inequality (Steele, 2004). Combining the*
506    *derivations of Eq. (A.5) and Eq. (A.8) into Eq. (A.4), we obtain the final inequality as follows*

$$\mathop{\mathbb{E}}_{S\sim\mathbb{P}^N}\left[\left(\widetilde{R}(\theta,\omega) - R(\theta,\omega)\right)^2\right] \leq \frac{1}{N}\mathop{\mathbb{V}}_{S\sim\mathbb{P}^N}\left[\mathbf{p}^{\mathrm{t}}(\mathbf{x})^\top \mathbf{l}\big(g_\omega(f_\theta(\mathbf{x}))\big)\right] + \mathcal{O}\left(\mathbb{E}_{\mathbf{x}}\left[\|\mathbf{p}^{\mathrm{t}}(\mathbf{x}) - \mathbf{p}^*(\mathbf{x})\|_2\right]\right)^2 \tag{A.9}$$

507    ## A.4 PROOF OF THEOREM 1

508    **Theorem 1** *Define* $\Delta(K) = \min\left\|\mathbf{p}^{\mathrm{t}}(\mathbf{x}_i) - \mathbf{p}^*(\mathbf{x}_i)\right\|_2 = \min\left\|\sum_{k=1}^{K}\lambda_\gamma(k,i)\widetilde{\mathbf{h}}_i^{(k)} - \mathbf{p}^*(\mathbf{x}_i)\right\|_2$ *with*
509    $K(K\geq 1)$ *given teachers, then we have (1)* $\Delta(K+1) \leq \Delta(K)$*, and (2)* $\lim_{K\to\infty}\Delta(K) = 0$*.*

510    **Proof.** Let us simplify the symbol $\lambda_\gamma(k,i)$ to $\lambda_k$ and consider the case with $K$ teachers, we have

$$\{\lambda_k^*\}_{k=1}^K = \arg\min_{\{\lambda_k\}_{k=1}^K} \left\|\sum_{k=1}^{K}\lambda_k\widetilde{\mathbf{h}}_i^{(k)} - \mathbf{p}^*(x)\right\|$$

$$\Delta\mathbf{p}_K = \sum_{k=1}^{K}\lambda_k^*\widetilde{\mathbf{h}}_i^{(k)} - \mathbf{p}^*(x),\ \Delta(K) = \|\Delta\mathbf{p}_K\|_2 \tag{A.10}$$

511    Next, let's consider the case with $(K+1)$ teachers, as follows

$$\Delta(K+1) = \min_{\{\lambda_k\}_{k=1}^K} \left\|\sum_{k=1}^{K+1}\lambda_k\widetilde{\mathbf{h}}_i^{(k)} - \mathbf{p}^*(\mathbf{x}_i)\right\|_2$$

$$\leq \min_{\lambda_{K+1}} \left\|\sum_{k=1}^{K}\lambda_k^*\widetilde{\mathbf{h}}_i^{(k)} + \lambda_{K+1}\widetilde{\mathbf{h}}_i^{(K+1)} - \mathbf{p}^*(\mathbf{x}_i)\right\|_2$$

$$= \min_{\lambda_{K+1}} \left\|\lambda_{K+1}\widetilde{\mathbf{h}}_i^{(K+1)} + \Delta\mathbf{p}_K\right\|_2 \tag{A.11}$$

$$\leq \sin\left(arc\cos\frac{<\Delta\mathbf{p}_K, \widetilde{\mathbf{h}}_i^{(K+1)}>}{\|\Delta\mathbf{p}_K\|_2 \cdot \|\widetilde{\mathbf{h}}_i^{(K+1)}\|_2}\right) \cdot \|\Delta\mathbf{p}_K\|_2$$

$$\leq \|\Delta\mathbf{p}_K\|_2 = \Delta(K)$$

512    where the equality in the fourth row of Eq. (A.11) holds under the condition that

$$\lambda_{k+1} = -\frac{<\Delta\mathbf{p}_K, \widetilde{\mathbf{h}}_i^{(K+1)}>}{\|\Delta\mathbf{p}_K\|_2 \cdot \|\widetilde{\mathbf{h}}_i^{(K+1)}\|_2} \cdot \frac{\|\Delta\mathbf{p}_K\|_2}{\|\widetilde{\mathbf{h}}_i^{(K+1)}\|_2} = -\frac{<\Delta\mathbf{p}_K, \widetilde{\mathbf{h}}_i^{(K+1)}>}{\|\widetilde{\mathbf{h}}_i^{(K+1)}\|_2^2} \tag{A.12}$$

Let $K \geq 2$ be the number of teachers, and the results of the $K$-th iteration can be defined as follows:

$$
\begin{aligned}
\Delta(K) &\leq \sin\left(arc\cos\frac{<\Delta\mathbf{p}_{K-1}, \widetilde{\mathbf{h}}_i^{(K)}>}{\|\Delta\mathbf{p}_{K-1}\|_2 \cdot \|\widetilde{\mathbf{h}}_i^{(K)}\|_2}\right) \cdot \Delta(K-1) \\
&\leq = \sin\left(arc\cos\frac{<\Delta\mathbf{p}_{K-1}, \widetilde{\mathbf{h}}_i^{(K)}>}{\|\Delta\mathbf{p}_{K-1}\|_2 \cdot \|\widetilde{\mathbf{h}}_i^{(K)}\|_2}\right) \cdot \sin\left(arc\cos\frac{<\Delta\mathbf{p}_{K-2}\widetilde{\mathbf{h}}_i^{(K-1)}>}{\|\Delta\mathbf{p}_{K-2}\|_2 \cdot \|\widetilde{\mathbf{h}}_i^{(K-1)}\|_2}\right) \cdot \Delta(K-2) \\
&\leq \cdots \\
&\leq \prod_{k=2}^{K} \sin\left(arc\cos\frac{<\Delta\mathbf{p}_{k-1}, \widetilde{\mathbf{h}}_i^{(k)}>}{\|\Delta\mathbf{p}_{k-1}\|_2 \cdot \|\widetilde{\mathbf{h}}_i^{(k)}\|_2}\right) \cdot \Delta(1)
\end{aligned}
$$

Since $\sin\left(arc\cos\frac{<\Delta\mathbf{p}_{k-1}, \widetilde{\mathbf{h}}_i^{(k)}>}{\|\Delta\mathbf{p}_{k-1}\|_2 \cdot \|\widetilde{\mathbf{h}}_i^{(k)}\|_2}\right) \leq 1$, and the equality holds when and only when $\Delta\mathbf{p}_{k-1}$ and $\widetilde{\mathbf{h}}_i^{(k)}$ are orthogonal, which in practice is hard to be satisfied, we have $\lim_{K\to\infty} \Delta(K) = 0$.

## A.5 Pseudo Code of MGSSL

The pseudo-code of the proposed MGSSL framework is summarized in Algorithm 1.

---

**Algorithm 1** Algorithm for the *Multi-teacher Knowledge Distillation* framework for MGSSL

---

**Input:** Graph $\mathcal{G} = (\mathcal{V}, \mathcal{E}, \mathbf{X})$, Number of Pretext Tasks: $K$, and Number of Epochs: $T$.
**Output:** Predicted Labels $\mathcal{Y}_U$, GNN Enocder $f_\theta(\cdot)$, and Prediction Head $g_\omega(\cdot)$.
1: Randomly initialize the parameters of $K$ teacher models and a student model.
2: Pre-train each teacher with individual task by Eq. (3) to get pre-trained parameters $\{\theta_k^*, \omega_k^*\}_{k=1}^K$.
3: **for** $t \in \{0, 1, \cdots, T-1\}$ **do**
4:     Output logits $\{\mathbf{h}_i^{(k)} = g_{\omega_k^*}(f_{\theta_k^*}(\mathcal{G}, i))\}_{k=1}^K$ from the pre-trained teachers and freeze them.
5:     Integrate the knowledge of different teachers by $\mathbf{p}^{\dagger}(\mathbf{x}_i) = \sum_{k=1}^K \lambda_\gamma(k, i)\sigma(\mathbf{h}_i^{(k)}/\tau)$.
6:     Jointly perform distillation by Eq. (4) and optimize the function $\lambda_\gamma(\cdot, \cdot)$ with loss $\mathcal{L}_W$.
7: **end for**
8: **return** Predicted labels $\mathcal{Y}_U$, GNN encoder $f_\theta(\cdot)$, and prediction head $g_\omega(\cdot)$.

---

## A.6 Dataset Statistics

*Eight* publicly available graph datasets are used to evaluate the proposed MGSSL framework. An overview summary of the statistical characteristics of datasets is given in Table. A1. For the three small-scale datasets, namely Cora, Citeseer, and Pubmed, we follow the data splitting strategy by Kipf & Welling (2016). For the four large-scale datasets, namely Coauthor-CS, Coauthor-Physics, Amazon-Photo, and Amazon-Computers, we follow Zhang et al. (2021); Luo et al. (2021) to randomly split the data into train/val/test sets, and each random seed corresponds to a different splitting. For the ogbn-arxiv dataset, we use the public data splits provided by the authors (Hu et al., 2020).

Table A1: Statistical information of the datasets.

| Dataset | Cora | Citeseer | Pubmed | Photo | CS | Physics | Computers | ogbn-arxiv |
|---|---|---|---|---|---|---|---|---|
| # Nodes | 2708 | 3327 | 19717 | 7650 | 18333 | 34493 | 13752 | 169343 |
| # Edges | 5278 | 4614 | 44324 | 119081 | 81894 | 247962 | 245861 | 1166243 |
| # Features | 1433 | 3703 | 500 | 745 | 6805 | 8415 | 767 | 128 |
| # Classes | 7 | 6 | 3 | 8 | 15 | 5 | 10 | 40 |
| Label Rate | 5.2% | 3.6% | 0.3% | 2.1% | 1.6% | 0.3% | 1.5% | 53.7% |

## A.7 HYPERPARAMETER SETTINGS

The following hyperparameters are set the same for all datasets: Adam optimizer with learning rate $lr$ = 0.01 (0.001 for `ogb-arxiv`) and weight decay $w$ = 5e-4; Epoch $E$ = 500; Layer number $L$ = 1 (2 for `ogb-arxiv`). The other dataset-specific hyperparameters are determined by an AutoML toolkit NNI with the hyperparameter search spaces as: hidden dimension $F = \{32, 64, 128, 256, 512\}$; distillation temperature $\tau = \{1, 1.2, 1.5, 2, 3, 4, 5\}$, and loss weights $\alpha, \beta = \{0.1, 0.5, 1, 5, 10, 20, 30\}$. For a fairer comparison, the model with the highest validation accuracy is selected for testing. Besides, the best hyperparameter choices for each dataset are available in the supplementary. Moreover, the experiments on both baselines and our approach are implemented based on the standard implementation in the DGL library (Wang et al., 2019) using the PyTorch 1.6.0 with Intel(R) Xeon(R) Gold 6240R @ 2.40GHz CPU and NVIDIA V100 GPU.

## A.8 DETAILS ON FIVE PRETEXT TASKS

In this paper, we evaluate the capability of `MGSSL` in automatic pretext tasks combinatorial search with five classical pretext tasks, including PAR (You et al., 2020b), CLU (You et al., 2020b), DGI (Velickovic et al., 2019), PAIRDIS (Jin et al., 2020), and PAIRSIM (Jin et al., 2020). Our motivations for selecting these five pretext tasks are 4-fold: (1) *Fair comparison.* To make a fair comparison with previous methods (e.g., AutoSSL), we keep in line with it in the setting of pretext tasks, i.e., using the same pool of pretext tasks. (2) *Simple but classical.* We should pick those pretext tasks that are simple but classical enough, rather than those that are overly complex, not time-tested, and not well known. This is to avoid, whether the resulting performance gains come from our proposed `MGSSL` or from the complexity of the selected pretext task itself, becoming incomprehensible and hard to explain. (3) *Comprehensive.* Different pretext tasks implicitly involve different inductive biases, so it is important to consider different aspects comprehensively when selecting pretext tasks, rather than picking too many homogeneous and similar tasks. (4) *Applicability.* There is no conflict at all between Graph SSL automation and designing more powerful pretext tasks; as a general framework, `MGSSL` is applicable to other more complex self-supervised tasks. However, the focus of this paper is on the knowledge distillation framework rather than on the specific task design, and it is also impractical to enumerate all existing graph SSL methods in a limited space.

**PAR and CLU.** The pretext task of Node Clustering (CLU) pre-assigns a pseudo-label $\widehat{y}_i$, e.g., the cluster index, to each node $v_i \in \mathcal{V}$ by $K$-means clustering algorithm (MacQueen, 1965). The learning objective of this pretext task can then be formulated as a classification problem, as follows

$$\mathcal{L}_{\text{ssl}}\left(\theta, \eta\right) = \frac{1}{N} \sum_{v_i \in \mathcal{V}} \ell\Big(g_\eta(f_\theta(\mathcal{G}, i), \widehat{y}_i\Big) \tag{A.13}$$

When node attributes are not available, another choice to obtain pseudo-labels is based on the topology of the graph structure. Specifically, graph partitioning (PAR) predicts partition pseudo-labels obtained by the Metis graph partition (Karypis & Kumar, 1998). While CLU and PAR are very similar, they extract ***feature-level*** and ***topology-level*** knowledge from the graph, respectively. A key hyperparameter of them is the category number of pseudo-labels #P, which is set to #P=10 for CLU and #P=400 (100 for Amazon-Photo and Amazon-Computers, 1000 for Citeseer) for PAR, following the settings by Jin et al. (2021). In practice, CLU can be easily extended to other variants by adopting other data clustering algorithms (Wu et al., 2022c;b).

**DGI.** Deep Graph Infomax (DGI) is proposed to contrast the node representations and corresponding high-level summary of graphs. First, it applies an augmentation transformation $\mathcal{T}(\cdot)$ to obtain an augmented graph $\widetilde{\mathcal{G}} = \mathcal{T}(\mathcal{G})$. Then a shared graph encoder $f_\theta(\cdot)$ is applied to obtain node embeddings $\mathbf{h}_i = f_\theta(\mathcal{G}, i)$ and $\widetilde{\mathbf{h}}_i = f_\theta(\widetilde{\mathcal{G}}, i)$. Besides, a global mean pooling is applied to obtain the graph-level representation $\mathbf{h}_{\widetilde{g}} = \frac{1}{N} \sum_{i=1}^{N} \widetilde{\mathbf{h}}_i$. Finally, the learning objective is defined as follows

$$\mathcal{L}_{\text{ssl}}(\theta) = -\frac{1}{N} \sum_{v_i \in \mathcal{V}} \mathcal{MI}\left(\mathbf{h}_{\widetilde{g}}, \mathbf{h}_i\right) \tag{A.14}$$

where $\mathcal{MI}(\cdot, \cdot)$ is the InfoNCE mutual information estimator (Gutmann & Hyvärinen, 2010), where the negative samples to contrast with $\mathbf{h}_{\widetilde{g}}$ is $\{\mathbf{h}_j\}_{j \neq i}$. The pretext task of DGI extracts knowledge at

the graph level. To improve the computational efficiency for large-scale graphs, we will randomly sample 2000 nodes to contrast the representations between these sampled nodes and the whole graph.

**PAIRDIS.** The pretext task of PAIRDIS aims to guide the model to preserve ***global topology information*** by predicting the shortest path length between nodes. It first randomly samples a certain amount of node pairs $\mathcal{S}$ and calculates the pairwise node shortest path length $d_{i,j} = d(v_i, v_j)$ for node pairs $(v_i, v_j) \in \mathcal{S}$. Furthermore, it groups the shortest path lengths into four categories: $C_{i,j} = 0, C_{i,j} = 1, C_{i,j} = 2$, and $C_{i,j} = 3$ corresponding to $d_{i,j} = 1, d_{i,j} = 2, d_{i,j} = 3$, and $d_{i,j} \geq 4$, respectively. The learning objective can be formulated as a multi-class classification problem,

$$\mathcal{L}_{\text{ssl}}(\theta, \eta) = \frac{1}{|\mathcal{S}|} \sum_{(v_i, v_i) \in \mathcal{S}} \ell\Big(g_\eta\big(|f_\theta(\mathcal{G})_{v_i} - f_\theta(\mathcal{G})_{v_j}|\big), C_{i,j}\Big) \tag{A.15}$$

where $\ell(\cdot)$ denotes the cross entropy loss and $g_\eta(\cdot)$ linearly maps the input to a 1-dimension value. A key hyperparameter in PAIRDIS is the size of $\mathcal{S}$, which is set to $|\mathcal{S}| = 400$ for all eight datasets.

**PAIRSIM.** Unlike PAIRDIS, which focuses on the global topology, PAIRSIM adopts link prediction as a pretext task to predict feature similarities between node pairs and thus capture ***local connectivity information*** from the graph. PAIRSIM first masks $m$ edges $\mathcal{M} \in \mathcal{E}$ and also samples $m$ edges $\overline{\mathcal{M}} \in \{(v_i, v_j)|v_i, v_j \in \mathcal{V} \text{ and } (v_i, v_j) \notin \mathcal{E}\}$. Then, the learning objective of PAIRSIM is to predict whether there exists a link between a given node pair, which can be formulated as follows

$$\mathcal{L}_{\text{ssl}}(\theta, \eta) = \frac{1}{2m}\Big(\sum_{e_{i,j} \in \mathcal{M}} \ell\big(g_\eta(|f_\theta(\mathcal{G}, i) - f_\theta(\mathcal{G}, j)|), 1\big) + \sum_{e_{i,j} \in \overline{\mathcal{M}}} \ell\big(g_\eta(|f_\theta(\mathcal{G}, i) - f_\theta(\mathcal{G}, j)|), 0\big)\Big)$$

where $\ell(\cdot)$ denotes the cross entropy and $g_\eta(\cdot)$ linearly maps the input to a 1-dimension value. The task of PAIRSIM aims to help the GNN model learn more local structural information. A key hyperparameter in PAIRSIM is the size of $\mathcal{M}$, which is set to $|\mathcal{M}| = 400$ by default for all datasets.

## A.9 DISCUSSION ON RELATED WORK

**Graph Self-supervised Learning (SSL).** The primary goal of graph SSL is to learn transferable knowledge from unlabeled data through well-designed pretext tasks. There have been hundreds of SSL pretext tasks proposed in the past few years. For example, DSSL (Xiao et al., 2022) performs self-supervised learning on non-homophilous graphs, which can leverage both useful local structure and global semantic information. Besides, Kim et al. (2022) proposes a Discrepancy-based Self-supervised LeArning (D-SLA) framework that aims to learn the exact discrepancy between the original and the perturbed graphs by using a discriminator. Moreover, a recent SSL work, GraphAME (Hou et al., 2022) proposes a masked autoencoder that extends masked modeling to graphs by performing masked feature reconstruction and re-mask decoding. We refer interested readers to the recent surveys (Wu et al., 2021; Xie et al., 2021; Liu et al., 2021) for more information. Despite the great success, these methods mostly focus on designing more powerful but complex self-supervised pretext tasks, with little effort to explore how to leverage multiple existing tasks more efficiently.

**Automated Machine Learning.** One of the most related topics to us is the automated loss function search (Zhao et al., 2021; Weber et al., 2020; Hutter et al., 2019; Waring et al., 2020; Yao et al., 2018). However, most of these methods are specifically designed for image data and may not be applicable to graph-structured data. For example, the loss function of PAIRDIS involves two nodes, which is hardly compatible with the node-specific loss function of PAR. A recent work JOAO (You et al., 2021) on graph contrastive learning is proposed to automatically select data augmentation, but it is tailored for graph classification and single-task contrastive learning and is difficult to extend to multi-task self-supervised learning. Another related work is AUX-TS (Han et al., 2021), which adaptively combines different auxiliary tasks in order to generalize to other tasks during the fine-tuning stage of transfer learning, which is hard to extend directly to the graph self-supervised learning setting. Besides, BGNN (Guo et al., 2022) proposes a novel adaptive knowledge distillation framework to sequentially transfer knowledge from multiple GNNs into a student GNN. However, their main contribution is to sequentially enhance GNN representation learning in an adaptive and "boosting" manner, rather than learning to weigh multiple different teachers at the same time as done in our work. Moreover, DMTGAT (Wang et al.) formulates GNN architecture search as a bi-level multi-objective optimization problem (BL-MOP) to find a set of Pareto architectures and their Pareto

weights. However, the above works (Guo et al., 2022; Wang et al.) has little to do with the topic of our work, i.e., multi-task graph self-supervised learning. A recent work, ParetoGNN (Ju et al., 2022), is very close to our work. ParetoGNN is simultaneously self-supervised by multiple pretext tasks, which are dynamically reconciled to promote the Pareto optimality during pre-training, such that the graph encoder actively learns knowledge from every pretext task while minimizing potential conflicts. Another closest work, AutoSSL (Jin et al., 2021), formulates the automated self-supervised task search as a bi-level optimization problem and solves it via meta-gradient descent.

**Graph Knowledge Distillation.** Recent years have witnessed the great success of graph knowledge distillation in learning graph representations. Several previous works on graph distillation try to distill knowledge from large teacher GNNs to smaller student GNNs, termed as GNN-to-GNN knowledge distillation (KD) (Zhang et al., 2020; Ren et al., 2021). For example, KDGA (Wu et al., 2022a) investigates how to distill knowledge from the augmented graph to the original graph to address distributional shifts. The other branch of graph knowledge distillation is to directly distill from teacher GNNs to lightweight student MLPs, termed GNN-to-MLP KD. For example, GLNN (Zhang et al., 2021) directly distills knowledge from teacher GNNs to vanilla MLPs by imposing KL-divergence between their logits. Besides, FF-G2M (Wu et al., 2023a) propose to factorize GNN knowledge into low- and high-frequency components in the spectral domain and propose a novel framework to distill both low- and high-frequency knowledge from teacher GNNs into student MLPs. Moreover, RKD (Wu et al., 2023b) quantifies the reliability of knowledge for reliable knowledge distillation. Despite the great progress made, none of the above knowledge distillation works have anything to do with self-supervised learning. The main purpose of these efforts is to distill knowledge from GNNs to lightweight GNN or MLP, not involving either knowledge integration or multi-teacher KD.

## A.10 RESULTS FOR GRAPH CLASSIFICATION AND VISION TASKS

To further evaluate how well MGSSL works on other graph-related tasks, we consider three classic graph-related tasks, including graph regression, node clustering, and link prediction. In terms of the task of **graph regression**, we report in Table. A2 the performance (ROC-AUC) of five classical pretext tasks (e.g., AttrMask, ContextPred, GPT-GNN, GraphCL, and Graph LoG) for the molecular property prediction task on 8 molecular datasets. Besides, we evaluate the performance of Loss Weighting and MGSSL-TS in the multi-tasking setting. Note that AutoSSL and ParetoGNN are not included in the comparison since they are not applicable to graph-level regression tasks. From the results in Table. A2, it can be seen that MGSSL-TS performs better than all single-task models and outperforms Loss Weighting by a wide margin. In addition, we take FeatRec, TopoRec, RepDecor, MI-NG, and MI-NSG as pretext tasks and compare the performance of AutoSSL, ParetoGNN, and MGSSL-TS on **node clustering** and **link prediction** tasks, which are measured by the NMI and AUC metrics, respectively. The reported results in Table. A3 (node clustering) and Table. A4 (link prediction) also demonstrate the superiority of MGSSL-TS over AutoSSL and ParetoGNN.

Table A2: Performance (ROC-AUC, %) comparison of the five baseline teachers, vanilla Loss Weighting, and MGSSL-TS for the *graph-level task* of molecular property prediction. The arrows indicate whether the two methods improve relative to the average performance of the five baselines.

| Method | BACE | BBBP | ClinTox | SIDER | Tox21 | Toxcast | MUV | HIV | Avg. Rank |
|---|---|---|---|---|---|---|---|---|---|
| AttrMask | 77.4±0.2 | 65.3±1.6 | 70.3±7.5 | 55.1±0.7 | 74.4±0.5 | 62.6±0.1 | 75.4±2.7 | 75.9±0.4 | 4.50 |
| ContextPred | 77.3±1.0 | 69.0±2.0 | 66.9±7.6 | 58.7±1.6 | 72.9±0.8 | 61.7±0.7 | 73.6±0.3 | 76.1±2.4 | 4.63 |
| GPT-GNN | 78.6±2.9 | 65.3±1.5 | 56.1±8.9 | 57.9±0.2 | 74.3±0.7 | 63.3±0.3 | 75.6±1.8 | 74.8±1.4 | 4.25 |
| GraphCL | 77.5±1.6 | 69.9±1.6 | 72.1±4.7 | 59.9±1.5 | 75.1±0.8 | 62.8±0.7 | 75.1±1.5 | 74.5±0.6 | 3.13 |
| GraphLoG | 78.1±1.0 | 66.4±2.8 | 64.1±3.4 | 59.5±2.4 | 73.9±1.4 | 62.3±0.6 | 73.5±1.0 | 75.5±0.5 | 5.00 |
| Loss Weighting | 76.5±0.7 ↓ | 67.2±1.2 ↑ | 62.8±6.0 ↓ | 56.4±1.3 ↓ | 72.6±1.0 ↓ | 60.4±1.2 ↓ | 74.2±2.1 ↓ | 76.6±1.5 ↑ | 5.38 |
| MGSSL-TS | 79.7±1.4 ↑ | 70.8±1.5 ↑ | 73.5±4.5 ↑ | 60.7±1.7 ↑ | 74.7±1.2 ↑ | 64.4±0.9 ↑ | 76.4±1.9 ↑ | 78.2±1.0 ↑ | 1.13 |

Furthermore, we evaluate the applicability of MGSSL to image data by considering three classical vision tasks, including image classification (evaluated by Recall@5) on ImageNet, object category detection (evaluated by mAP) on PASCAL VOC 2007, and depth prediction (% Pixels below 1.25) on NYU v2. Four different classical visual pretext tasks are taken into account, including Relative Position, Colorization, Exemplar Nets, and Motion Segmentation (Doersch & Zisserman, 2017). To adapt MGSSL to vision tasks, we conduct graph construction by taking images as nodes and con-

Table A3: Performance (NMI) comparison of five (single-task) teachers and three multi-tasking methods for the *node clustering* task, where **bold** and underline denote the best and second metrics.

| Method | Wiki-CS | Pubmed | AM-Photo | AM-Computers | Co-CS | Co-Physics | Avg. Rank |
|---|---|---|---|---|---|---|---|
| FeatRec | 43.04±1.92 | 30.24±0.01 | 63.25±1.41 | 43.83±1.30 | 74.61±1.04 | 37.83±0.01 | 5.33 |
| TopoRec | 36.06±1.25 | 19.22±0.02 | 66.27±1.06 | 48.51±1.68 | 69.83±0.45 | 48.15±0.22 | 6.00 |
| RepDecor | 34.96±0.59 | 26.51±0.33 | 61.28±1.31 | 49.78±1.02 | 66.53±1.63 | 47.65±0.70 | 6.17 |
| MI-NG | 39.78±0.24 | 24.70±0.61 | 65.32±1.57 | 48.78±0.56 | 66.16±0.62 | 49.98±0.54 | 5.50 |
| MI-NSG | 47.77±0.14 | 24.34±0.01 | 55.92±1.01 | 49.61±0.55 | 74.91±0.82 | 56.83±0.01 | 4.50 |
| AutoSSL | 36.99±0.21 | 28.99±0.26 | 64.06±0.65 | 41.85±0.36 | 74.04±0.22 | 55.23±0.18 | 5.33 |
| ParetoGNN | 47.52±0.29 | **34.74±0.06** | 68.25±1.25 | 52.53±0.34 | 74.94±0.98 | 60.43±0.13 | 2.00 |
| MGSSL-TS | **48.20±1.47** | 32.82±0.28 | **69.49±1.37** | **53.20±0.49** | **76.10±1.18** | **61.51±0.23** | 1.17 |

Table A4: Performance (AUC) comparison of five (single-task) teachers and three multi-tasking methods for the *link prediction* task, where **bold** and underline denote the best and second metrics.

| Method | Wiki-CS | Pubmed | AM-Photo | AM-Computers | Co-CS | Co-Physics | Avg. Rank |
|---|---|---|---|---|---|---|---|
| FeatRec | 95.79±0.05 | 93.96±0.05 | 95.47±0.15 | 90.51±0.17 | 96.51±0.02 | 95.97±0.06 | 4.67 |
| TopoRec | 92.69±0.25 | 94.17±0.94 | 95.13±1.25 | 95.89±0.12 | 96.43±0.37 | 97.98±0.01 | 4.67 |
| RepDecor | 93.64±0.09 | 87.55±0.06 | 94.86±0.16 | 86.45±0.57 | 94.00±0.16 | 96.48±0.08 | 6.67 |
| MI-NG | 92.48±0.08 | 91.48±0.17 | 95.33±0.05 | 94.19±0.04 | 97.83±0.11 | 90.18±0.15 | 5.67 |
| MI-NSG | 95.90±0.04 | 92.22±0.02 | 95.22±0.64 | 94.11±0.07 | 92.13±0.01 | 93.13±0.06 | 5.67 |
| AutoSSL | 93.86±0.02 | 86.84±1.30 | 95.57±0.13 | 93.99±0.03 | 95.71±0.15 | 95.93±0.07 | 5.67 |
| ParetoGNN | 96.48±0.01 | 94.58±0.02 | 96.08±0.08 | **97.16±0.04** | **98.18±0.02** | 98.33±0.03 | 1.67 |
| MGSSL-TS | **96.89±0.02** | **95.26±0.24** | **96.76±0.15** | 96.80±0.11 | 97.77±0.08 | **98.50±0.06** | 1.33 |

necting the k-Nearest Neighbors (kNN) of each image to build edges. As can be seen from the experimental results in Table. A5, the MGSSL-TS can consistently outperform each of the individual tasks as well as Loss Weighting across three visual tasks and datasets. Furthermore, we have also provided the results of constructing the graph by thresholding, where two images with cosine similarity greater than 0.7 will be connected by an edge. The results in Table. A5 show that constructing the graph by kNN outperforms thresholding, and we speculate that this is because kNN guarantees the balance of node degrees in the constructed graph and prevents the over-squeezing problem that is common in graph learning. Note that we provide preliminary results on three graph-related tasks and three vision tasks only to demonstrate the potential of the proposed MGSSL framework for handling general multi-task self-supervised learning, and deeper exploration will be left for future work.

Table A5: Performance comparisons on three classical visual tasks, including image classification on ImageNet, object category detection on PASCAL VOC, and depth prediction on NYU v2.

| Graph Construction | Method | ImageNet | PASCAL | NYU |
|---|---|---|---|---|
| | | Recall@5 | mAP | % Pixels below 1.25 |
| - | Relative Position | 59.2 | 66.8 | 80.5 |
| | Colorization | 62.1 | 65.5 | 71.8 |
| | Exemplar Nets | 53.4 | 60.1 | 71.3 |
| | Motion Segmentation | 60.9 | 64.5 | 74.6 |
| k-Nearest Neighbor | Loss Weighting | 65.3 | 63.8 | 78.3 |
| | MGSSL-TS | **69.4** | **73.2** | **81.7** |
| Thresholding | Loss Weighting | 64.5 | 64.3 | 76.8 |
| | MGSSL-TS | 67.7 | 70.5 | 79.5 |

## A.11 APPLICABILITY TO DIFFERENT GNN ARCHITECTURES

We report the performance of Vanilla, AutoSSL, and MGSSL-TS on five large-scale datasets (CS, Physics, Photo, Computers, and ogbn-arxiv) under the JT setting, respectively. Table. A6 shows that our MGSSL-TS works well for all three classic GNN architectures, especially with GATs, where MGSSL significantly outperforms the previous important baseline, AutoSSL, by a large margin.

Table A6: Comparison of the applicability to three GNN architectures on five datasets.

| GNN Architecture | Method | CS | Physics | Photo | Computers | ogbn-arxiv |
|---|---|---|---|---|---|---|
| GCNs | Vanilla | 92.16 | 93.94 | 91.52 | 86.58 | 70.94 |
| GCNs | AutoSSL | **93.54** | 95.10 | 92.94 | 88.72 | 72.26 |
| GCNs | MGSSL-TS | 93.46 | **95.54** | **94.22** | **89.72** | **72.72** |
| GATs | Vanilla | 91.86 | 93.58 | 91.76 | 86.74 | 70.74 |
| GATs | AutoSSL | 92.80 | 94.82 | 93.04 | 88.46 | 71.96 |
| GATs | MGSSL-TS | **93.70** | **95.76** | **94.18** | **89.88** | **72.84** |
| GrapgSAGE | Vanilla | 92.30 | 93.86 | 91.80 | 86.50 | 70.80 |
| GrapgSAGE | AutoSSL | 93.28 | 95.14 | 93.16 | 88.68 | 72.10 |
| GrapgSAGE | MGSSL-TS | **93.52** | **95.48** | **94.30** | **89.64** | **72.66** |

## A.12 PERFORMANCE IN THE P&F SETTING

We compare MGSSL-TS with several representative graph SSL baselines under the P&F setting in Table. A7, where we present the performance improvement of MGSSL-TS over AutoSSL and ParetoGNN. As you can see, MGSSL also has significant advantages under the P&F setting.

Table A7: Performance comparison with classical self-supervised baselines in the P&F setting, where **bold** and underline denote the best and second metrics on each dataset, respectively.

| Method | Cora | Citeseer | Pubmed | CS | Physics | Photo | Computers |
|---|---|---|---|---|---|---|---|
| GCNs | 81.72 | 71.48 | 79.26 | 91.04 | 93.06 | 91.90 | 86.36 |
| DGI | 82.30 | 71.80 | 76.80 | 91.39 | 93.42 | 92.11 | 87.19 |
| GMI | 83.00 | 72.40 | 79.90 | 91.46 | 93.60 | 92.22 | 87.43 |
| MVGRL | 82.90 | 72.60 | 79.40 | 91.69 | 93.79 | 92.50 | 87.89 |
| GRACE | 80.00 | 71.70 | 79.50 | 91.21 | 93.12 | 92.01 | 86.83 |
| GCA | 82.86 | 72.64 | 79.78 | 91.84 | 93.80 | 92.40 | 87.95 |
| BGRL | 83.48 | 72.81 | 80.30 | 92.10 | 94.24 | 92.89 | 88.28 |
| AutoSSL | 82.96 | 72.76 | 80.14 | **92.48** | 93.88 | 92.36 | 88.00 |
| ParetoGNN | 83.34 | 72.98 | 79.95 | 92.24 | 94.43 | 92.78 | 88.14 |
| MGSSL-LF | 84.22 | 73.58 | **80.62** | 92.36 | 94.80 | 93.32 | **88.68** |
| MGSSL-TS | **84.38** | **73.70** | 80.54 | 91.94 | **94.96** | **93.52** | 88.42 |
| ΔAutoSSL | +1.71% | +1.29% | +0.60% | -0.13% | +1.15% | +1.26% | +0.48% |
| ΔParetoGNN | +1.25% | +0.99% | +0.84% | +0.13% | +0.56% | +0.80% | +0.32% |

## A.13 DETAILS ON EXPERIMENTAL RESULTS

Table. A8 provides the numerical values of results in Fig. 8(a) and Fig. 8(b). The settings of five teacher combinations are (1) one teacher: PAR; (2) two teachers: PAR and CLU; (3) three teachers: PAR, CLU, and DGI; (4) four teachers: PAR, CLU, DGI, and PAIRDIS; and (5) five teachers: PAR, CLU, DGI, PAIRDIS, and PAIRSIM. As shown in Table. A9, MGSSL-LF and MGSSL-TS always perform better than other heuristic methods; more importantly, their performance increases consistently with the number of teachers, reaching the best at a number of five teachers.

Table A8: Ablation study on knowledge integration under different number of teachers, where **bold** and underline denote the best and second metrics for each teacher number, respectively. The best performance (i.e., the optimal teacher number) for each integration scheme is marked in blue.

| Method | Citeseer | | | | | Coauthor-CS | | | | |
|---|---|---|---|---|---|---|---|---|---|---|
| | 1 (+PAR) | 2 (+CLU) | 3 (+DGI) | 4 (+PAIRDIS) | 5 (+PAIRSIM) | 1 (+PAR) | 2 (+CLU) | 3 (+DGI) | 4 (+PAIRDIS) | 5 (+PAIRSIM) |
| Random | $72.72_{\pm 0.36}$ | $72.96_{\pm 0.47}$ | $72.66_{\pm 0.39}$ | $72.94_{\pm 0.43}$ | $72.86_{\pm 0.45}$ | $92.30_{\pm 0.67}$ | $92.12_{\pm 0.54}$ | $92.68_{\pm 0.47}$ | $92.38_{\pm 0.53}$ | $92.26_{\pm 0.64}$ |
| Average | $72.72_{\pm 0.36}$ | $73.04_{\pm 0.34}$ | $72.92_{\pm 0.42}$ | $73.16_{\pm 0.39}$ | $73.26_{\pm 0.37}$ | $92.30_{\pm 0.67}$ | $92.36_{\pm 0.49}$ | $92.48_{\pm 0.60}$ | $93.02_{\pm 0.55}$ | $93.16_{\pm 0.47}$ |
| Weighted | $72.72_{\pm 0.36}$ | $73.16_{\pm 0.32}$ | $73.24_{\pm 0.46}$ | $73.40_{\pm 0.43}$ | $73.36_{\pm 0.40}$ | $92.30_{\pm 0.67}$ | $92.52_{\pm 0.46}$ | $92.90_{\pm 0.51}$ | $92.76_{\pm 0.39}$ | $93.28_{\pm 0.53}$ |
| MGSSL-LF | $72.72_{\pm 0.36}$ | $\mathbf{73.56}_{\pm 0.39}$ | $\mathbf{73.68}_{\pm 0.33}$ | $\mathbf{74.18}_{\pm 0.40}$ | $\mathbf{74.34}_{\pm 0.31}$ | $92.30_{\pm 0.67}$ | $\underline{92.64}_{\pm 0.44}$ | $\mathbf{93.30}_{\pm 0.37}$ | $\mathbf{93.68}_{\pm 0.52}$ | $\mathbf{93.86}_{\pm 0.36}$ |
| MGSSL-TS | $72.72_{\pm 0.36}$ | $\underline{73.24}_{\pm 0.44}$ | $\underline{73.34}_{\pm 0.40}$ | $\underline{73.58}_{\pm 0.38}$ | $\underline{74.20}_{\pm 0.42}$ | $92.30_{\pm 0.67}$ | $\mathbf{92.88}_{\pm 0.34}$ | $\underline{93.04}_{\pm 0.26}$ | $\underline{93.38}_{\pm 0.31}$ | $\underline{93.46}_{\pm 0.25}$ |

## A.14 RESULTS ON MORE TEACHERS

We conduct an ablation study of knowledge integration with more numbers of teachers in Fig. A1, which also takes into account those SOTA SSL baselines in Table. 2. It can be seen that MGSSL-TS can consistently benefit from more teachers and outperform the Average and Weighted integrations, especially with a larger number of teachers. However, as the number of teachers increases, the performance improvements may eventually reach a theoretical maximum, as bounded by Theorem. 1.

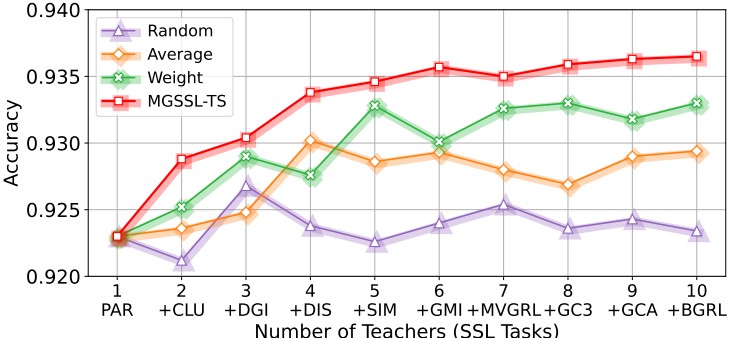

Figure A1: Ablation study on knowledge integration under different numbers of teachers (pretext tasks), where MGSSL-TS can consistently benefit from more teachers, outperforming the *Average* and *Weighted* integration, especially with a larger number of teachers. However, as the number of teachers increases, the performance gains reach a theoretical maximum, as bounded by Theorem. 1.

## A.15 TIME COMPLEXITY AND COMPUTATIONAL EFFICIENCY

The time complexity of MGSSL mainly comes from three parts: (1) Teacher Training $\mathcal{O}(K(|\mathcal{V}|dF + |\mathcal{E}|F))$; (2) Knowledge Integration $\mathcal{O}(K|\mathcal{V}|F)$; and (3) Knowledge Distillation $\mathcal{O}(|\mathcal{V}|F)$, where $d$ and $F$ are the dimensions of input and hidden spaces. The total time complexity $\mathcal{O}(K(|\mathcal{V}|dF + |\mathcal{E}|F))$ is linear w.r.t the number of nodes $|\mathcal{V}|$ and edges $|\mathcal{E}|$, and the number of teachers (SSL tasks) $K$. In practice, $K$ is usually less than 10, and more importantly, we can reduce the complexity of Teacher Training from $\mathcal{O}(K(|\mathcal{V}|dF + |\mathcal{E}|F))$ to $\mathcal{O}((|\mathcal{V}|dF + |\mathcal{E}|F))$ by parallelizing the training of multiple teachers on hardware devices such as GPUs. We compare the training time of MGSSL with the joint training (JOINT-T) of multiple pretext tasks with fixed loss weights in Table. A9. It can be seen that while MGSSL needs to train multiple teacher models separately, it still has advantages over JOINT-T in terms of training time, mainly because: (1) each teacher in MGSSL can be trained **in parallel**, which greatly reduces the time expense; (2) the training with multiple tasks is more difficult to optimize than the training with one single task, so each training epoch of JOINT-T takes longer time than MGSSL; and (3) JOINT-T is more difficult to converge with higher complexity, i.e., it requires more training epochs to converge. Instead, MGSSL takes much less time for each model, resulting in less overall training time. (4) MGSSL-LF and MGSSL-TS differ only in their knowledge integration schemes, so their overall training time is very close and much less than JOINT-T.

Table A9: Comparison of the computational costs (training time) of three methods on nine datasets.

| Method | Cora | Citeseer | Pubmed | CS | Physics | Photo | Computers | ogbn-arxiv | ogbn-products |
|--------|------|----------|--------|-----|---------|-------|-----------|-----------|---------------|
| JOINT-T | 17.87s | 18.57s | 75.18s | 98.83s | 171.61s | 36.73s | 51.90s | 1362.28s | $6.41 \times 10^4$s |
| MGSSL-LF | 15.42s | 16.09s | 68.31s | 91.76s | 158.73s | 32.61s | 45.96s | 1289.73s | $5.89 \times 10^4$s |
| MGSSL-TS | 15.53s | 16.22s | 68.67s | 92.14s | 159.24s | 32.84s | 46.26s | 1294.67s | $5.96 \times 10^4$s |

## A.16 DISTILLED KNOWLEDGE ANALYSIS FROM A FREQUENCY PERSPECTIVE

We follow previous work (Wu et al., 2023a) in decomposing knowledge into high- and low-frequency components, which are measured by mean cosine similarity and KL-divergence, respec-

tively. The low-frequency knowledge of five teacher models and the student model is measured by the *mean cosine similarity* of nodes with their 1-order neighbors. In addition, the high-frequency knowledge is measured by the *KL-divergence* between the pairwise distances of five teacher models with the student model. See Appendix D of Wu et al. (2023a) for details on how to measure high-/low- frequency knowledge. We provide a comparison of high- and low-frequency knowledge for the five teacher and student models in Fig. A2, from which it can be observed that (1) low-frequency knowledge (a.k.a., common knowledge) from multiple teachers can be well learned by the student, and (2) the student model learns high-frequency knowledge differently from each teacher. A smaller KL-divergence metric indicates better distillation of high-frequency knowledge from the teacher.

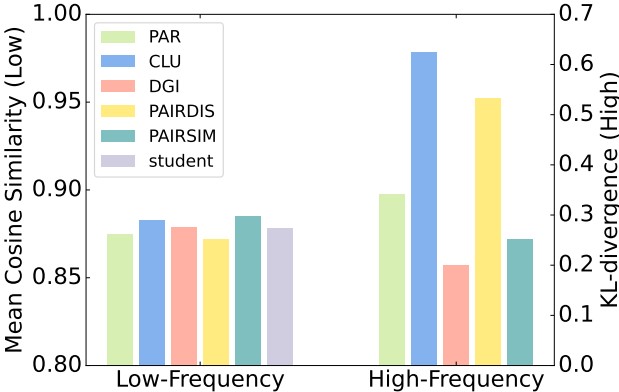

Figure A2: Low-/High- frequency Knowledge analysis on the Cora dataset. The low-frequency knowledge of five teachers and one student is measured by *mean cosine similarity*. The high-frequency knowledge is measured by *KL-divergence* between five teachers with the student model.

## A.17 SYMBOL TABLE

Unless particularly specified, the symbols used in this paper are illustrated in Table. A10.

Table A10: Symbols used in this paper.

| Symbols | Descriptions |
|---|---|
| $\mathbb{R}^m$ | $m$-dimensional Euclidean space |
| $x, \mathbf{x}, \mathbf{X}$ | Scalar, vector, matrix |
| $\mathcal{G}$ | A graph $g = (\mathcal{V}, \mathcal{E}, \mathbf{X})$ |
| $\mathcal{V}$ | Node set in the graph $\mathcal{G}$ |
| $\mathcal{E}$ | Edge set in the graph $\mathcal{G}$ |
| $\mathbf{X}$ | Node feature matrix in the graph $\mathcal{G}$ |
| $N$ | Number of nodes in the graph $\mathcal{G}$ |
| $(\mathcal{V}_L, \mathcal{Y}_L)$ | Labeled set of nodes and labels |
| $(\mathcal{V}_U, \mathcal{Y}_U)$ | Unlabeled set of nodes and labels |
| $f_\theta(\cdot)$ | GNN encoder of the student model |
| $g_\eta(\cdot)$ | Prediction head of the student model |
| $f_{\theta_k}(\cdot)$ | GNN encoder of the $k$-th teacher model |
| $g_{\eta_k}(\cdot)$ | Prediction head of the $k$-th teacher model |
| $f_{\theta_k^*}(\cdot)$ | Pre-trained GNN encoder of the $k$-th teacher model |
| $g_{\eta_k^*}(\cdot)$ | Pre-trained prediction head of the $k$-th teacher model |
| $\mathcal{L}_{\text{task}}(\theta, \omega)$ | loss of downstream task |
| $\mathcal{L}_{\text{ssl}}^{(k)}(\theta_k, \omega_k)$ | loss of the $k$-th SSL pretext task |
| $\lambda_k$ | loss weight of the $k$-th SSL pretext task |
| $\lambda_\gamma(\cdot, \cdot)$ | weighting function parameterized by $\gamma$ |
| $\mathbf{x}_i$ | input node feature of node $v_i$ |
| $\mathbf{z}_i$ | output logit of node $v_i$ in the student model |
| $\mathbf{h}_i^{(k)}$ | output logit of node $v_i$ in the $k$-th teacher model |
| $\widetilde{\mathbf{z}}_i = \sigma(\mathbf{z}_i/\tau)$ | activated logit of node $v_i$ in the student model |
| $\widetilde{\mathbf{h}}_i^{(k)} = \sigma(\mathbf{h}_i^{(k)}/\tau)$ | activated logit of node $v_i$ in the $k$-th teacher model |
| $R(\theta, \omega)$ | Bayesian objective |
| $\mathbf{p}^*(\mathbf{x})$ | Bayesian class-probability |
| $\widetilde{R}(\theta, \omega)$ | Distillation objective |
| $\mathbf{p}^{\text{t}}(\mathbf{x}_i) \doteq \sum_{k=1}^K \lambda_\gamma(k, i)\widetilde{\mathbf{h}}_i^{(k)}$ | Integrated teacher probability |
| $y_i$ | Ground-Truth label of node $v_i$ |
| $\mathbf{e}_{y_i}^\top$ | One-hot label of node $v_i$ |
| $\ell(\cdot, \cdot)$ | Cross-entropy loss |
| $\mathcal{L}_{KL}(\cdot, \cdot)$ | KL-divergence loss |
| $C$ | Number of category |
| $K$ | Number of SSL pretext tasks (teachers) |
| $\tau$ | Temperature coefficient |
| $\alpha, \beta$ | Loss weights |
| $\theta, \eta, \gamma, \boldsymbol{\mu}_k, \boldsymbol{\nu}, \mathbf{W}$ | Learnable model parameters |

