# OpenReview forum: "An Instance-Level Framework for Multi-tasking Graph Self-Supervised Learning"
_ICLR.cc/2024/Conference — Submitted to ICLR 2024_

### Official Review · Reviewer_YwHq · 2023-10-29

**Soundness:** 3 good
**Presentation:** 3 good
**Contribution:** 3 good
**Rating:** 6
**Confidence:** 5

**Summary:**

This paper designs a multi-tasking graph self-supervised learning framework to improve downstream graph learning performance. Different from the conventional multi-tasking SSL, it proposes a multi-teacher KD framework with an instance-level knowledge integration module, with the parameterized knowledge distillation module, which can output instance-wise knowledge combination weights. The proposed approach suggests a significant performance improvement over other single-pretext tasks, e.g., vanilla MTL, AutoSSL, and ParetoGNN.

**Strengths:**

1. This paper is well-written and provides sufficient empirical observations of combining multiple self-supervised tasks.
2. The experiment for node classification is conducted over a broad datasets and the improvement is significant.
3. Some theoretic results can be directly supported by the empirical results.
4. The proposed method with weighing different teachers for each node is novel.

**Weaknesses:**

1. The main concern is the experiments in this paper are only related to the classification task in a multi-tasking setting, while the proposed method is task-irrelevant. Although the some vision tasks are involved in the appendix, the graph-related tasks, such as node clustering and link prediction, should be investigated.
2. The Eq. (4) is quite misleading. The proposed distillation is in an offline fashion, and the optimality of teacher has already in the objective. As a result, the constraint is a little bit unnecessary. Also, the parameters for optimization are not clearly presented in the objective.

**Questions:**

1. In page 2, you claim “Secondly, balancing multiple tasks by loss weighting during the pre-training phase makes it hard to scale the pre-trained model to emerging tasks. To incorporate new tasks, it requires to re-pretrain a new model from scratch.” Is it an over- pessimistic claim for the existing methods? Which may implicit exaggerate the contribution of the proposed method.
2. How do you perform vision task in Appendix 10? Please provide more details for the graph construction. Maybe you can discuss the performance difference between different methods to build the graph.
3. Is there more evidence for Figure 4? From Guidance 1, the integrated teacher probability is as close as possible to the true Bayesian probability. What is the specific reason for the failure of the other two schemes?

---

> ### Author Response · Authors · 2023-11-14
> **Response to Reviewer YwHq**
>
> Thanks for your insightful comments and constructive suggestions! We mark the contents that already existed (but were missed by reviewers) in red, and those revised or newly added contents in blue in the revised manuscript. Next, we will address the concerns you raised, one by one, as described below:
>
> ---
>
> **Q1**: The main concern is the experiments in this paper are only related to the classification task in a multi-tasking setting, while the proposed method is task-irrelevant. Although the some vision tasks are involved in the appendix, the graph-related tasks, such as node clustering and link prediction, should be investigated.
>
> **A1**: We have added performance comparisons of MGSSL, AutoSSL, and ParetoGNN on various graph-related downstream tasks (graph regression, node clustering, link prediction) in **Tables A2/A3/A4** and **Lines 642-654** in **Appendix A.10**. These experiments demonstrate the applicability of MGSSL to various downstream tasks and its superiority over AutoSSL and ParetoGNN.
>
> ---
>
> **Q2**: The Eq. (4) is quite misleading. The proposed distillation is in an offline fashion, and the optimality of teacher has already in the objective. As a result, the constraint is a little bit unnecessary. Also, the parameters for optimization are not clearly presented in the objective.
>
> **A2**: The parameters to be optimized in Eq. (4) during KD are the student model ($\{\theta, \omega\}$) and the weighting function $\lambda_\gamma(k,i)$ (parameterized by $\gamma$). Therefore, although each teacher model is frozen before KD, the integrated teacher (the optimality of teacher) $\sum_{k=1}^K\lambda_\gamma(k,i) \widetilde{\mathbf{h}}\_i^{(k)}$ changes as $\lambda_\gamma(k,i)$ is updated during KD. As a result, Eq. (4) is essentially more like an online KD, and perhaps we have not made it clear what the parameters of optimization are, leading you to misinterpret it as "an offline fashion". To make it clear, we have added the above explanation to **Lines 127-130** of the revised manuscript.
>
> ---
>
> **Q3**: In page 2, you claim “Secondly, balancing multiple tasks by loss weighting during the pre-training phase makes it hard to scale the pre-trained model to emerging tasks. To incorporate new tasks, it requires to re-pretrain a new model from scratch.” Is it an over- pessimistic claim for the existing methods? Which may implicit exaggerate the contribution of the proposed method.
>
> **A3**: Thanks for your insightful comment. We have modified the previous expression (somewhat over-pessimistic),
>
> > "To incorporate new tasks, it requires to re-pretrain a new model from scratch."
> >
>
> to a more moderate expression,
>
> > "To incorporate new tasks, it requires to re-weight the losses of new tasks and existing tasks to re-pretrain the model." in the revised manuscript.”
> >
>
> ---
>
> **Q4**: How do you perform vision task in Appendix 10? Please provide more details for the graph construction. Maybe you can discuss the performance difference between different methods to build the graph.
>
> **A4**: To adapt MGSSL to vision tasks, we conduct graph construction by taking images as nodes and connecting the k-Nearest Neighbors (kNN) of each image to build edges. As you suggested, we have also provided the results of constructing the graph by thresholding in **Table A5** of **Appendix A.10**, where two images with cosine similarity greater than 0.7 will be connected by an edge. The results show that constructing the graph by kNN outperforms thresholding, and we speculate that this is because kNN guarantees the balance of node degrees in the constructed graph and prevents the over-squeezing problem that is common in graph learning. We have added the above explanation to **Lines 659-669** of the revised manuscript.
>
> ---
>
> **Q5**: Is there more evidence for Figure 4? From Guidance 1, the integrated teacher probability is as close as possible to the true Bayesian probability. What is the specific reason for the failure of the other two schemes?
>
> **A5**: Among the three schemes in Figure 4, only "Adaptive (ours)" follows Guideline 1, while "Averaged" and "Weighted" are two heuristic (vanilla) schemes that have nothing to do with Guideline 1. The reasons for the failure of "Averaged" and "Weighted" are that they cannot differentiate important teachers from irrelevant ones, and may mislead the student in the presence of low-quality teachers, as the manuscript described. By contrast, "Adaptive (ours)" follows Guideline 1, which enables the integrated teacher probability to be as close as possible to the true Bayesian probability by adaptive teacher weighting. To make it clear, we have added the above explanation to **Lines 186-193** of the revised manuscript.
>
> ---
>
> In light of these responses, we hope we have addressed your concerns. If we have left any notable points of concern unaddressed, please do share and we will attend to these points. We sincerely hope that you can appreciate our efforts on responses and revisions and raise your score.

---

> ### Author Response · Authors · 2023-11-21
> **Looking forward to your reply**
>
> Dear Reviewer,
>
> Thank you for your insightful and helpful comments once again. We greatly appreciate your feedback. We have carefully responded to each of your questions point-by-point.
>
> Unlike previous years, there will be no second stage of author-reviewer discussions this year. As the deadline for the discussion is approaching, we kindly request you to inform us if you have any additional questions.
>
> Best regards,
>
> Authors.

---

> > ### Comment · Reviewer_YwHq · 2023-11-22
> >
> > Thanks for your response. The final decision will be released soon, after the discussion with other reviewers and AC.

---

> ### Author Response · Authors · 2023-11-22
>
> Thanks again for your insightful review and keeping a positive score for acceptance. If there is anything else that needs clarification, please don't hesitate to contact us!
>
> Best regards, Authors.

---

### Official Review · Reviewer_UNJF · 2023-10-31

**Soundness:** 3 good
**Presentation:** 2 fair
**Contribution:** 2 fair
**Rating:** 5
**Confidence:** 3

**Summary:**

This paper presents the MGSSL, which is a a novel multi-teacher knowledge distillation framework for instance-level Multi-tasking Graph SSL. This framework conducts knowledge integration in the fine-tuning phase, distilling the knowledge from multi-teacher models to the student models, which achieves an instance-level integration module. The authors provided sufficient theory to prove this framework can benefit from a wider range of teachers, and the experimental part also achieved competitive results.

**Strengths:**

My opinions towards the strengths of this papers are:
   + Originality. This paper proposes the idea of integrating knowledge in the fine-tuning phase, and thus achieved a breakthrough at the instance-level requirements for the first time.
   + Quality. This paper has clear method description, sufficient experiments, and competitive results.
   + Clarity. The figures of the methods plus theoretical proof clearly expressed the method of the paper.
   + Significance. A framework theoretical guideline brings new ideas to solve Graph Self-Supervised Learning problems.

**Weaknesses:**

First of all, the authors say that they integrate knowledge from multiple teacher models to the student models. I wonder what knowledge is integrated into the student models. I think the knowledge may be the common knowledge from the multiple teacher models. The authors may present more analysis.

Second, I find the paper not easy to follow. The symbols maybe complex for me to understand.  As a result, I suggest the authors to simplify and well structure the used symbols.

**Questions:**

My two concerns are as the Weaknesses part. Pls refer to Weaknesses.

---

> ### Author Response · Authors · 2023-11-14
> **Response to Reviewer UNJF**
>
> Thanks for your insightful comments and constructive suggestions! We mark the contents that already existed (but were missed by reviewers) in red, and those revised or newly added contents in blue in the revised manuscript. Next, we will address the concerns you raised, one by one, as described below:
>
> ---
>
> **Q1**: I wonder what knowledge is integrated into the student models. I think the knowledge may be the common knowledge from the multiple teacher models. The authors may present more analysis.
>
> **A1**: We follow previous work [1] in decomposing knowledge into high- and low- frequency components, which are measured by cosine similarity and KL-divergence, respectively (see Appendix D of [1] for details on how to measure high/low frequency knowledge). We provide a comparison of high- and low-frequency knowledge for the five teacher and student models in **Figure A2** and **Lines 711-721** of **Appendix A.16**, from which it can be observed that (1) low-frequency knowledge (a.k.a., common knowledge) from multiple teachers can be well learned by the student, and (2) the student model differently learn high-frequency knowledge from each teacher.
>
> [1] Wu L, Lin H, et al. Extracting Low-/High-Frequency Knowledge from Graph Neural Networks and Injecting it into MLPs: An Effective GNN-to-MLP Distillation Framework[J]. AAAI, 2023.
>
> ---
>
> **Q2**: I find the paper not easy to follow. The symbols maybe complex for me to understand. As a result, I suggest the authors to simplify and well structure the used symbols.
>
> **A2**: Thanks for your constructive suggestions. We have simplified and reorganized the symbols used in this paper and provided a list of symbols in **Table A10** of **Appendix A.17**. We hope this helps you understand and appreciate our contributions. If anything is still unclear, please do not hesitate to contact us.
>
> ---
>
> In light of these responses, we hope we have addressed your concerns. If we have left any notable points of concern unaddressed, please do share and we will attend to these points. We sincerely hope that you can appreciate our efforts on responses and revisions and thus raise your score.

---

> ### Author Response · Authors · 2023-11-21
> **Looking forward to your reply**
>
> Dear Reviewer,
>
> Thank you for your insightful and helpful comments once again. We greatly appreciate your feedback. We have carefully responded to each of your questions point-by-point.
>
> Unlike previous years, there will be no second stage of author-reviewer discussions this year. As the deadline for the discussion is approaching, we kindly request you to inform us if you have any additional questions.
>
> Best regards,
>
> Authors.

---

### Official Review · Reviewer_RhiN · 2023-11-02

**Soundness:** 3 good
**Presentation:** 3 good
**Contribution:** 2 fair
**Rating:** 5
**Confidence:** 3

**Summary:**

This paper proposes a novel approach in graph self-supervised learning called Multi-teacher Knowledge Distillation Framework for Instance-level Multitasking Graph Self-Supervised Learning (MGSSL). MGSSL uses multiple teachers for
different tasks, integrating their knowledge for each instance, and distills it into a student model. This method shifts from loss weighting in pre-training to knowledge integration in fine-tuning, allowing compatibility with numerous tasks without retraining from scratch. Theoretical justifications are provided, and extensive experiments show that MGSSL's performance is comparable to state-of-the-art methods by combining simple classical tasks.

**Strengths:**

+ The proposed multi-teacher knowledge distillation framework for instance-level multitasking graph self-supervised learning (MGSSL) is a novel approach. It addresses the limitations of existing methods by focusing on instance-level requirements and scalability to new tasks.

+ The paper successfully identifies and tackles key challenges in the field: the need for localized combinations of tasks for different instances, poor scalability to emerging tasks, and the lack of theoretical guarantees for performance improvement with more tasks.

+ Providing theoretical justification for the potential of MGSSL to benefit from a wider range of teachers (tasks) adds credibility and depth to  the research.

+ The extensive experiments and the resulting performance being comparable to state-of-the-art competitors lend strong empirical support to the proposed method.

**Weaknesses:**

- An evaluation of the scalability and computational costs of the proposed framework, especially in large-scale settings, would be a valuable addition.

- The author only compares their work with AutoSSL in Table 1. Since AutoSSL is a work from 2021, I suggest that the author should compare their work with more recent studies. This would further enhance the quality of the paper.

**Questions:**

Please see my comments in Weaknesses.

---

> ### Author Response · Authors · 2023-11-14
> **Response to Reviewer RhiN**
>
> Thanks for your insightful comments and constructive suggestions! We mark the contents that already existed (but were missed by reviewers) in red, and those revised or newly added contents in blue in the revised manuscript. Next, we will address the concerns you raised, one by one, as described below:
>
> ---
>
> **Q1**: An evaluation of the scalability and computational costs of the proposed framework, especially in large-scale settings, would be a valuable addition.
>
> **A1**: The discussion and analysis on the computational complexity and overhead of MGSSL has been provided in **Lines 694-709**, **Appendix A.15** of the original manuscript. Did you miss this part of the experiment? Following your suggestion, we have added the computational cost of MGSSL on a larger ogbn-products dataset (consisting of 2,449,029 nodes and 61,859,140 edges) to demonstrate its excellent scalability in **Table A9** in the revised manuscript.
>
> ---
>
> **Q2**: .The author only compares their work with AutoSSL in Table 1. Since AutoSSL is a work from 2021, I suggest that the author should compare their work with more recent studies. This would further enhance the quality of the paper.
>
> **A2**: One latest work, ParetoGNN, published in ICLR'2023, has been included in the comparison in **Table 1**.  Additionally, we provide more comparisons of MGSSL with AutoSSL and ParetoGNN on various graph-related tasks in **Tables A2/A3/A4** and **Lines 642-654** in **Appendix A.10**.
>
> ---
>
> In light of these responses, we hope we have addressed your concerns. If we have left any notable points of concern unaddressed, please do share and we will attend to these points. We sincerely hope that you can appreciate our efforts on responses and revisions and thus raise your score.

---

> ### Author Response · Authors · 2023-11-21
> **Looking forward to your reply**
>
> Dear Reviewer,
>
> Thank you for your insightful and helpful comments once again. We greatly appreciate your feedback. We have carefully responded to each of your questions point-by-point.
>
> Unlike previous years, there will be no second stage of author-reviewer discussions this year. As the deadline for the discussion is approaching, we kindly request you to inform us if you have any additional questions.
>
> Best regards,
>
> Authors.

---

### Official Review · Reviewer_xyJe · 2023-11-02

**Soundness:** 3 good
**Presentation:** 3 good
**Contribution:** 2 fair
**Rating:** 5
**Confidence:** 4

**Summary:**

This paper focuses on the problem of graph self-supervised learning and presents a multi-teacher knowledge distillation framework. Specifically, it trains multiple teachers with different pretext tasks, then integrates the knowledge of different teachers for each instance separately by two parameterized knowledge integration schemes (MGSSL-TS and MGSSL-LF), and finally distills it into the student model.

**Strengths:**

1.	The paper is clearly motivated and well-written.
2.	Theoretical analysis is provided to show that the proposed method has the potential to benefit from more teachers.

**Weaknesses:**

1.	Experiments on other downstream tasks (e.g. node clustering, link prediction, and partition prediction as in AutoSSL and ParetoGNN) are missing.
2.	It seems better to add the discussion and comparison on related works about graph knowledge distillation [abc] and recent self-supervised graph learning [def].
a.	Quantifying the Knowledge in GNNs for Reliable Distillation into MLP, ICML2023.
b.	Extracting Low-/High- Frequency Knowledge from Graph Neural Networks and Injecting it into MLPs: An Effective GNN-to-MLP Distillation Framework, AAAI2023.
c.	Knowledge Distillation Improves Graph Structure Augmentation for Graph Neural Networks, NeurIPS2022.
d.	Decoupled Self-supervised Learning for Graphs, NeurIPS2022.
e.	Graph Self-Supervised Learning with Accurate Discrepancy Learning, NeurIPS2022.
f.	GraphMAE: Self-Supervised Masked Graph Autoencoders, KDD2022.
3.	Typo: In the line2 of page6, “…this paper proposes two…” is miswritten as “…this paper propose stwo…”.

**Questions:**

Please refer to the weaknesses.

---

> ### Author Response · Authors · 2023-11-14
> **Response to Reviewer xyJe**
>
> Thanks for your insightful comments and constructive suggestions! We mark the contents that already existed (but were missed by reviewers) in red, and those revised or newly added contents in blue in the revised manuscript. Next, we will address the concerns you raised, one by one, as described below:
>
> ---
>
> **Q1**: Experiments on other downstream tasks (e.g. node clustering, link prediction, and partition prediction as in AutoSSL and ParetoGNN) are missing.
>
> **A1**: We have added performance comparisons of MGSSL, AutoSSL, and ParetoGNN on various graph-related downstream tasks (graph regression, node clustering, link prediction) in **Tables A2/A3/A4** and **Lines 642-654** in **Appendix A.10**. These experiments demonstrate the applicability of MGSSL to various downstream tasks and its superiority over AutoSSL and ParetoGNN.
>
> ---
>
> **Q2**: It seems better to add the discussion and comparison on related works about graph knowledge distillation and recent self-supervised graph learning.
>
> **A2**: Thanks for your valuable suggestion. We have added the discussion on graph knowledge distillation and recent graph self-supervised learning in **Lines 591-602** and **Lines 626-640** of **Appendix A.9**, and the six papers you mentioned are all included, cited, and discussed. Moreover, we have included GraphMAE as an additional baseline in **Table 2**.
>
> ---
>
> **Q3**: Typo: In the line2 of page6, “…this paper proposes two…” is miswritten as “…this paper propose stwo…”.
>
> **A3**: All typos have been corrected in the revised manuscript.
>
> ---
>
> In light of these responses, we hope we have addressed your concerns. If we have left any notable points of concern unaddressed, please do share and we will attend to these points. We sincerely hope that you can appreciate our efforts on responses and revisions and thus raise your score.

---

> ### Author Response · Authors · 2023-11-21
> **Looking forward to your reply**
>
> Dear Reviewer,
>
> Thank you for your insightful and helpful comments once again. We greatly appreciate your feedback. We have carefully responded to each of your questions point-by-point.
>
> Unlike previous years, there will be no second stage of author-reviewer discussions this year. As the deadline for the discussion is approaching, we kindly request you to inform us if you have any additional questions.
>
> Best regards,
>
> Authors.

---

### Author Response · Authors · 2023-11-19
**Look forward to further feedback**

Dear Reviewers,

We would like to express our sincere gratitude for dedicating your time to reviewing our manuscript. Unlike previous years, there will be no second stage of author-reviewer discussions this year, and a recommendation needs to be provided by November 22, 2023. Considering that the date is approaching, we look forward to hearing from you! We have thoroughly considered all the feedback and provided a detailed response a few days ago. We hope that our response addresses your concerns to your satisfaction. If you still need any clarification or have any other questions, please do not hesitate to let us know.

Best regards,

Authors.

---

### Author Response · Authors · 2023-11-23
**Global Response**

Dear reviewers,

To address the concerns raised by the reviewers, the following revisions have been made and the revised manuscript has been uploaded.

- More experiments on graph-related tasks (graph regression, node clustering, link prediction) in **Tables A2/A3/A4** and **Lines 642-654** in **Appendix A.10**.

- More discussions on graph knowledge distillation and recent graph self-supervised learning in **Lines 591-602** and **Lines 626-640** of **Appendix A.9**.

- More comparisons with latest works in **Table 1** (ParetoGNN) and **Table 2** (GraphMAE).

- Analyzing which knowledge has been distilled from a frequency perspective in  in **Figure A2** and **Lines 711-721** of **Appendix A.16**.

Since we have not received replies to our rebuttal from all reviewers, we summarize the key points from their initial reviews as follows:

**Reason to Accept**

- A1. Provide theoretical justification to demonstrate the potential for MGSSL to benefit from more SSL tasks. (Reviewer xyJe, RhiN, UNJF, YwHq)

- A2. The paper is well-written with a strong and clear motivation, successfully identifies and tackles key challenges in the field for the first time. (Reviewer xyJe, RhiN, YwHq)

- A3. This paper makes a breakthrough in instance-level multi-tasking SSL, and weighing different teachers for each node is novel. (Reviewer RhiN, UNJF, YwHq)

- A4. Extensive experiments are conducted and the performance is comparable to SOTA competitors, which lends strong empirical support to the proposed method. (Reviewer RhiN, UNJF, YwHq)

**Reason to Reject**

- R1. Lack discussion and comparison on related works, such as Graph KD and recent Graph SSL. (Reviewer xyJe, RhiN)

- R2. More graph-related tasks, such as node clustering and link prediction, should be investigated. (Reviewer xyJe, YwHq)

- R3. More details need to be included, e.g., computational efficiency, notation, and explanations on Figure 4 and Eq. (4). (Reviewer RhiN, UNJF, YwHq)

We would like to clarify these concerns as follows:

- Regarding to R1, we have added the discussions on graph KD and recent graph SSL in **Lines 591-602** and **Lines 626-640** of **Appendix A.9**.

- Regarding to R2, we have added performance comparisons of MGSSL, AutoSSL, and ParetoGNN on various graph-related downstream tasks (graph regression, node clustering, link prediction) in **Tables A2/A3/A4** and **Lines 642-654** in **Appendix A.10**. Besides, we have added comparisons with latest works in **Table 1** (ParetoGNN) and **Table 2** (GraphMAE).

- Regarding to R3, the discussion and analysis on the computational complexity and overhead of MGSSL has been provided in **Lines 694-709**, **Appendix A.15**. A list of symbols is provided in **Table A10** of **Appendix A.17**. Answers to questions about Eq. (4) and Figure (4) are in the response to reviewer YwHq.

We sincerely hope that you can appreciate our efforts and provide a timely reply before the end of the author-reviewer discussion stage (November 22nd, AOE).

Best regards, Authors.

---

### Meta-Review · Area_Chair_dUQ8 · 2023-12-08

**Metareview:**

This is a borderline paper. A major concern lies in that the experimental results are not comprehensive enough to support the claim of the proposed method. In addition, the presentation of this work can be further improved to make the introduction of the proposed method clearer.

After the authors' rebuttal, the major concern remains. Therefore, this work is not ready for publication based on it current shape.

**Justification For Why Not Higher Score:**

A major concern remains.

**Justification For Why Not Lower Score:**

N/A

---

### Decision · Program_Chairs · 2024-01-16

Reject